# Quantifying information accumulation encoded in the dynamics of biochemical signaling

Ying Tang [1,2], Adewunmi Adelaja[1,2], Felix X.-F. Ye[3], Eric Deeds [1,4], Roy Wollman [1,4,5✉] & Alexander Hoffmann [1,2✉]

Cellular responses to environmental changes are encoded in the complex temporal patterns of signaling proteins. However, quantifying the accumulation of information over time to direct cellular decision-making remains an unsolved challenge. This is, in part, due to the combinatorial explosion of possible configurations that need to be evaluated for information in time-course measurements. Here, we develop a quantitative framework, based on inferred trajectory probabilities, to calculate the mutual information encoded in signaling dynamics while accounting for cell-cell variability. We use it to understand NFκB transcriptional dynamics in response to different immune threats, and reveal that some threats are distinguished faster than others. Our analyses also suggest specific temporal phases during which information distinguishing threats becomes available to immune response genes; one specific phase could be mapped to the functionality of the IκBα negative feedback circuit. The framework is generally applicable to single-cell time series measurements, and enables understanding how temporal regulatory codes transmit information over time.

---

[1] Institute for Quantitative and Computational Biosciences, University of California, Los Angeles, CA, USA. [2] Department of Microbiology, Immunology, and Molecular Genetics, University of California, Los Angeles, CA, USA. [3] Department of Applied Mathematics & Statistics, Johns Hopkins University, Baltimore, MD, USA. [4] Department of Integrative Biology and Physiology, University of California, Los Angeles, CA, USA. [5] Department of Chemistry and Biochemistry, University of California, Los Angeles, CA, USA. ✉email: rwollman@ucla.edu; ahoffmann@ucla.edu

Cells have the capacity to sense, respond, and adapt to their environment through biochemical signaling pathways[1–4], which convey external information to intracellular effectors that control cellular core functions[5] such as gene expression, metabolism, or cell shape and mobility. These signaling pathways respond to perturbations in the presence of nutrients, pathogens, cytokines, growth factors, toxins, or radiation that may function as stimuli or signaling inputs. Prior work has established that the temporal patterns (dynamics) of signaling molecular abundance within these pathways play a crucial role in determining the cellular response specific to each stimulus[6–8]. Moreover, signaling typically occurs over a timescale of hours, as it involves de novo protein synthesis and degradation. Regulatory circuits that control cellular core functions must interpret signaling dynamics when information about the identity and dose of the stimulus becomes available. Therefore, quantifying the temporal accumulation of information for stimuli-discrimination is essential to revealing how cells adapt to environmental changes and make cell fate decisions in real time.

Signaling through biochemical pathways is affected by the stochastic nature of molecular interactions[9,10] that may diminish its reliability as different individual cells of the same cell type receiving the same stimulus respond differently. To quantify information transmitted through noisy biochemical pathways, information-theoretic approaches have been adapted to analyze single-cell measurements of signaling molecules[11–17]. The mutual information (MI) between the stimulus conditions and the single-cell signaling responses is an estimate on the amount of information about the stimulus identity and dose that are encoded in the signaling molecular activity[18]. However, it remains an outstanding challenge to quantify the MI encoded in the dynamics of signaling and further to uncover how information accumulates over time, as cells react to environmental changes in real time with the information available at a certain timepoint[19–22]. That is because the possible configurations of response trajectories increase combinatorially with the number of timepoints, and this combinatorial explosion hinders an exact estimation on trajectory probabilities and MI.

Though several information measures consider two consecutive timepoints[23–26], none estimates the MI from an entire trajectory. One recently developed approach[27] allows estimates for several timepoints. However, it does not extract the information encoded in the dynamical patterns of responses, because it is not sensitive to the sequence of timepoints and does not distinguish between differentially permuted timepoints. Another method[28] analyzed the MI in dynamic signals using a machine learning decoder. However, such an analysis only provides a lower bound as it is unclear how much information might be lost in training the classifier. For example, classifiers employing linear principal components[29] may be inadequate to discriminate oscillatory and non-oscillatory trajectories.

Here, we address the complexity challenge of quantifying information accumulation from the trajectories with combinatorially increasing configurations. We employ a modeling approach to learn the ensemble of complex response trajectories using a time-inhomogeneous Markov model[30] or a hidden Markov model[31,32]. We then present a general workflow to identify the appropriate model and the optimal number of parameters for calculating the trajectory probability. This forms the basis for the dynamical mutual information (dMI), a measure we propose to quantify MI in the trajectory space.

We demonstrate the workflow by applying it to the signaling dynamics of the pathogen-response transcription factor NFκB, which shows complex oscillatory and non-oscillatory trajectories and determines the expression of immune response genes[22,33]. For our data of NFκB signaling in macrophages, we show that a hidden Markov model is superior to a Markov model and gives a posterior trajectory probability for each response trajectory. Then, the dMI framework reveals the capacity of cells to discriminate different stimulus conditions, and is capable of quantifying information accumulation with high temporal resolution unlike the previous methods. Specifically, the present method could reveal the timing of recognizing certain immune threats, and identify molecular mechanisms for specific temporal phases of information gain.

## Results

**Toward quantifying information accumulation encoded in the dynamics of biochemical signaling molecules.** Cellular signaling molecules often encode information about the biological condition via the temporal trajectories, or dynamics, of their activities (Fig. 1a, upper). The MI between categorical stimulus conditions and response trajectories quantifies the extent of stimulus-distinction over time (Fig. 1a, lower), and the maximum MI for certain stimulus conditions represents the maximum capacity of discriminating these conditions[18,27]. For example, $\log_2 M$ bits of maximum MI reveal perfect distinction on $M$ different stimulus conditions. The rate or temporal profile with which cells acquire the information to discriminate specific stimulus conditions remains largely unexplored.

Pioneering work in estimating the maximum MI[18] considered the data of single-cell signaling activity at a single timepoint as a distribution across cells (timepoint method), and calculated the MI among the distributions under various stimulus conditions (Fig. 1b, upper). When a timecourse of data is measured, it can be applied to each timepoint individually. However, this method does not take into account the information transmitted through the timecourse of signaling responses, and thus cannot reveal how MI accumulates while underestimating the capacity of the signaling channel.

To quantify the information from signal timecourses, a second approach[27] treated the response from each single cell as a multivariate vector (Fig. 1b, middle), and used density estimation (nearest neighbor estimator) to calculate the probability of every data vector (vector method). Though such treatment includes the information of the timecourse, it is not capable of distinguishing dynamical patterns in the time series, because a permutation on the ordering of data points does not alter the density estimation on the data vectors. Besides, the approach is also restricted to a few (<10) timepoints, because sampling the vectorial distribution is subject to a combinatorial explosion and becomes inaccurate when the number of timepoints increases.

To extract the maximum MI embedded in the timecourse of signaling responses, a decoding-based method[28] was developed using a machine learning classifier. However, training the classifier may involve an unknown amount of information loss, leading to the realization[28] that this method provides a lower bound on the true MI. Since it builds on the linear principal components of the time series data, it may not discriminate oscillatory dynamics well, and thus does not fully extract the information encoded in complex biological signaling dynamics. The statistical estimation[34], which uses linear regression to estimate the trajectory probabilities' ratio between stimuli, may also be inaccurate when applied to oscillatory trajectories.

Therefore, how to extract the full trajectory information or how the information accumulates over time remains an unsolved problem that requires the development of a new algorithmic framework (Fig. 1b, lower). Such a framework should be able to distinguish signaling activities with various dynamical patterns, including both oscillatory and non-oscillatory patterns. Thus, we reasoned that it requires a model that can adequately learn the ensemble of biological signaling trajectories under each stimulus

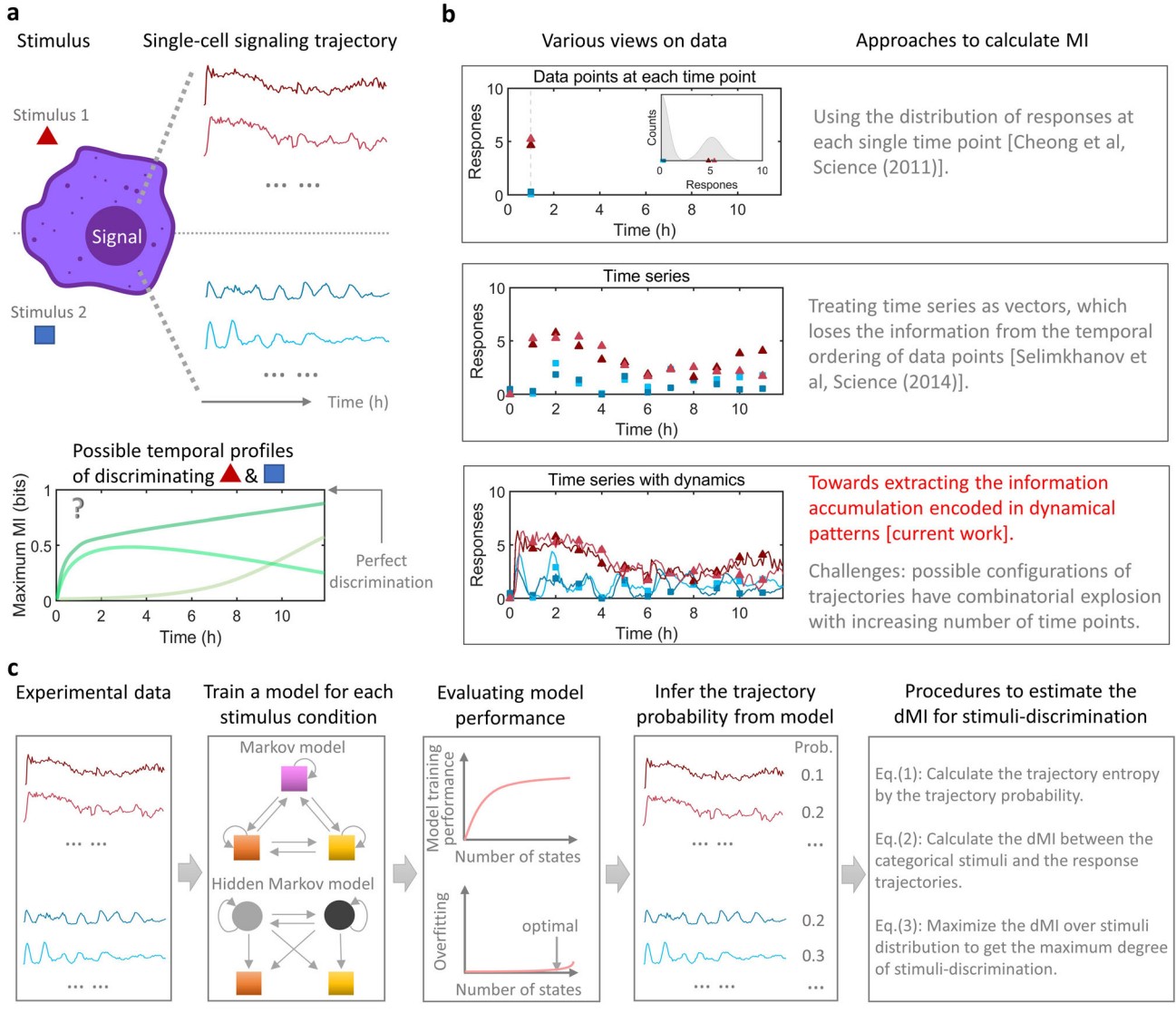

**Fig. 1 Toward quantifying information accumulation encoded in the dynamics of biochemical signaling molecules. a** (Upper) When cells encounter a stimulus, the responses of a major signaling molecule transmit information to the nucleus so that cells respond appropriately. The temporal trajectories of single-cell signaling response can have rich dynamics, such as the two exemplified trajectories under each separately added stimulus. (Lower) The maximum mutual information (MI) between the responses and the stimulus conditions quantifies the number of conditions that can be effectively distinguished. If the information is perfectly transmitted, it would reach $\log_2(M)$ bits for $M$ distinct stimulus conditions. The information transmits over time, and the temporal profile of cells acquiring the information for stimuli-discrimination remains to be explored. **b** An overview of the three representative approaches of quantifying the mutual information. **c** The workflow of the present approach. The measured trajectory ensemble under each stimulus condition can be learned by training a time-inhomogeneous Markov model or a hidden Markov model (circles are hidden states and squares are emission states in the model schematic). Under the optimal number of states, such stochastic models enable to infer the trajectory probabilities and develop the dynamical mutual information (dMI), which quantifies the mutual information encoded in signaling dynamics cumulatively.

condition. To account for cell-to-cell variable trajectories, we explored whether models rooted in statistical physics and machine learning[35,36], namely the time-inhomogeneous Markov model[30] or the hidden Markov model[31,32], could be used to recapitulate complex biological signaling dynamics.

In the newly devised workflow (Fig. 1c), the experimentally measured trajectory ensemble (with full trajectory length) under each stimulus condition may be used to train either a Markov or a hidden Markov model. Such dynamical models may account for all possible temporal trajectories including oscillatory patterns and data features that may be due to technical or biological noise. Thus, they can be used to classify the rich dynamical patterns observed in the experimental data, beyond what the methods based on linear regression can do.

The model-training step of the workflow also involves a careful investigation on the proper number of parameters, because too few parameters may not allow full representation of the complex trajectory ensemble but too many parameters can cause overfitting that leads to overestimates of MI. Thus, an important step in the workflow is to evaluate the model-training performance and the extent of overfitting. Training performance is evaluated by quantifying the similarity (such as by the Kullback–Leibler (KL)-divergence) between the measured trajectory ensemble under each stimulus condition and the simulated trajectory ensemble from the trained model (see the next section and Methods). Overfitting is guarded against by identifying the number of states that would result in misclassification of the heterogeneity within each condition.

When an appropriate model has been generated, the next step is to infer the probability of each trajectory and its corresponding trajectory entropy. Specifically, given a trained Markov or hidden Markov model under each stimulus condition, every trajectory's probability $p(y_{1:n})$, where $y_{1:n}$ denotes a trajectory from timepoint 1 to $n$, can be obtained[32] by sampling from the inferred transition matrix and emission matrix. Then, the trajectory entropy for each single trajectory is given by[37]:

$$H(y_{1:n}) = -\log_2 p(y_{1:n}). \tag{1}$$

Note that the trajectory entropy is for a single trajectory, rather than the average on the trajectory ensemble. The trajectory entropy was originally formulated for mesoscopic non-equilibrium systems[32], such as a colloidal particle in a viscous fluid. Here, the trajectory entropy is calculated by inferring a stochastic model, e.g., a hidden Markov model, from the data of biochemical signaling responses.

As detailed in Methods and Supplementary Note 1, the MI between the chosen $_M$ conditions of stimuli set ($S$) and the signaling responses set ($R_{1:n}$) is

$$I(R_{1:n}; S) = H(R_{1:n}) - H(R_{1:n}|S), \tag{2}$$

where $H(R_{1:n}|S)$ and $H(R_{1:n})$ are the conditional and unconditional trajectory entropy. Since the trajectory probability was generated from the dynamical model, the information embedded in dynamical patterns of signaling responses was extracted. We thus termed this quantity as dMI. We note that the dMI extends the MI calculation[27] by revealing the information encoded in dynamical patterns.

Considering the probability distribution of the $M$ stimulus conditions $\boldsymbol{q} = \{q_1, q_2, \ldots, q_M\}$, the maximum dMI is obtained by the maximization with respect to this probability distribution:

$$I_{\max}(R_{1:n}; S) = \max_{\boldsymbol{q}} I(R_{1:n}; S), \tag{3}$$

under the constraint of $q_1 + q_2 + \ldots + q_M = 1$ and $q_i \geq 0$. The maximization in $I_{\max}(R_{1:n}; S)$ is conducted at each timepoint, $n = 1, 2, 3, \ldots$, as a quantification on the maximum extent of distinguishing the $M$ stimulus conditions cumulatively up to that timepoint. A computational optimization over the distribution of stimulus conditions can be different from the genuine optimization conducted by the cell, because cells might not weight the distribution of stimuli as in the computation. Thus, $I_{\max}(R_{1:n}; S)$ is a maximum estimate on the degree of stimuli-discrimination at the timepoint $n$.

The proposed workflow estimates the maximum dMI, of which the value depends on the stimulus conditions under consideration. It approximates to the channel capacity of the signaling channel if an exhaustive number of perturbing conditions (stimuli) were employed experimentally. In practice, one can only sample a finite number of stimulus conditions in the continuous-concentration space, resulting in the maximum dMI that can be gleaned from the available datasets. In other words, if $M$ distinct conditions were employed, perfectly transmitted information would result in $\log_2 M$ bits. A smaller value than $\log_2 M$ bits implies that cells cannot fully discriminate the stimuli via the dynamics of the signaling molecule under consideration.

**Applying the framework to NFκB signaling dynamics.** To apply and evaluate the workflow, we studied the immune response transcription factor NFκB[7] which responds with diverse dynamics to different immune threats. We obtained a dataset of NFκB trajectories from macrophages exposed to the cytokine TNF and four different pathogen-associated molecular patterns, namely the bacterial cell wall components LPS and Pam3CSK, and the nucleic acids CpG and polyIC[38]. Responses to each were

tracked for 12 h. Various concentrations were used for the stimuli with responses dependent on concentrations (Fig. 2a and Supplementary Fig. S1). Applying our workflow, we found that the trained Markov and hidden Markov model can generate similar trajectory ensembles to the data (Fig. 2b and Supplementary Fig. S2).

We evaluated the performance of model training by using three measures (Fig. 2c and Supplementary Fig. S3): the relative KL-divergence; the false $k$-nearest neighbor probability, and the rescaled log-likelihood of test dataset (see Methods). The measures indicate that with a sufficient number of states the model can approximately reproduce the ensembles of single-cell NFκB trajectories. For example, the relative KL-divergence measuring the relative accuracy of the model reaches below 0.1 and the false $k$-nearest neighbor probability approaches 0.4, close to the optimal mixing probability 0.5. For the hidden Markov model, we explored how the numbers of hidden and emission states affect the training performance (Supplementary Fig. S4). Increasing the number of either hidden or emission states generally improves the model performance.

Both time-inhomogeneous Markov and hidden Markov models can be used to recapitulate cell-response trajectory data, each having advantages for specific datasets, such as p53[39] (Supplementary Fig. S6) and p38, JNK, and ERK[40] (Supplementary Fig. S7). The time-inhomogeneous Markov model performs better in capturing synchronized features when a perturbation is provided at an intermediate timepoint; alternatively, two hidden Markov models may be used for the two time windows before and after the perturbation (Supplementary Fig. S7). For datasets with oscillations at various frequencies, the hidden Markov model is more suitable, because it can describe trajectories with history dependence. In terms of calculating the dMI, the time-inhomogeneous model may be more sensitive to overfitting (Supplementary Fig. S8), and the hidden Markov model is superior in that regard, as demonstrated below.

To detect and quantify overfitting, we compared the capacity for distinguishing various stimulus conditions or the heterogeneity of response trajectories within each condition due to stochasticity. Specifically, for each stimulus condition, we split the trajectory ensemble into two subsets and calculated the maximum dMI between the two equally partitioned subsets (Supplementary Fig. S5). The maximum MI between the two subsets should be near zero when the model is not overfitted, because both are under the same stimulus condition. We repeated the calculation for every stimulus condition, and obtained the average maximum dMI between each pair of subsets. Then, the calculation was done with different numbers of discretized states for the two models, which determined the maximum number of allowable states that did not lead to overfitting. As described, overfitting is detected when the maximum dMI for two subsets of the same condition increases above zero. For the available datasets, the time-inhomogeneous Markov model becomes overfitted with more than only four states (Supplementary Fig. S8). For the hidden Markov model, we focused on the present NFκB dataset with sufficient number of trajectories, and our analysis defined a regime of around 60 hidden states and 30 emission states that can train a high-performance model without overfitting.

In navigating the trade-off between model performance and overfitting when choosing the number of states (Supplementary Fig. S5), the hidden Markov model may be more robust as it requires only two inferred matrices. It also does not have the Markovian assumption of being memoryless, and can therefore generate trajectories with history dependence, such as oscillations at various frequencies. These properties make the hidden Markov model more suitable than the time-inhomogeneous Markov model to quantify the dMI for the NFκB dataset.

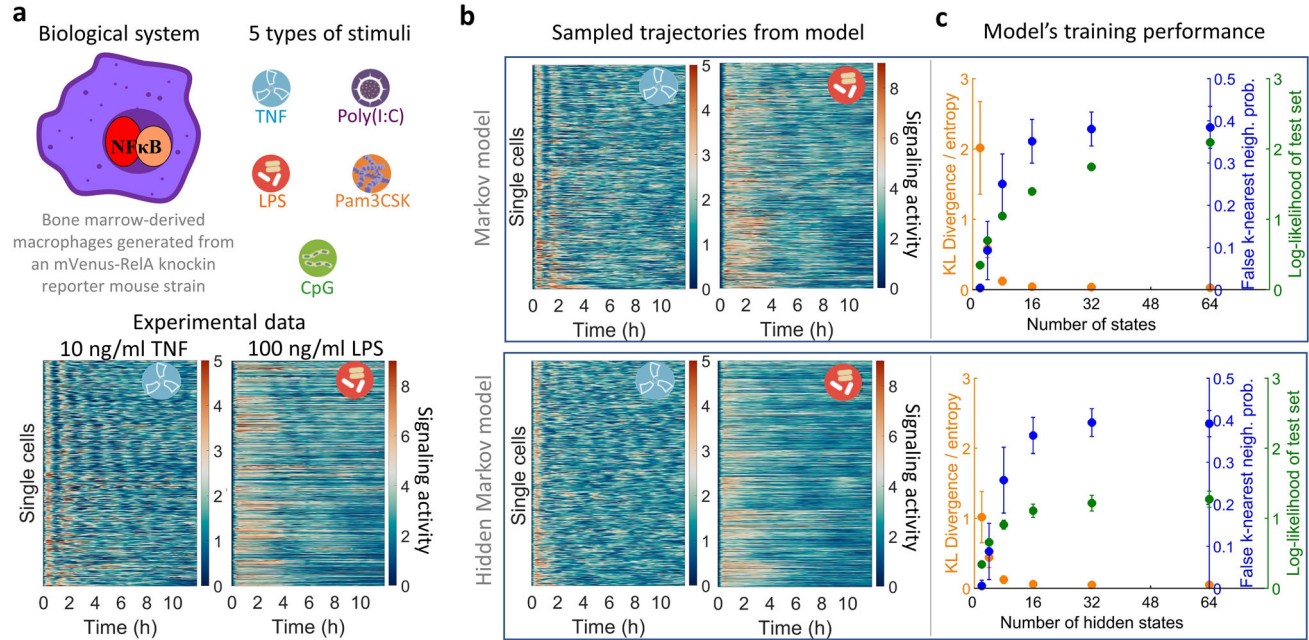

**Fig. 2 Applying the framework to NFκB signaling dynamics. a** When macrophages encounter a stimulus, the single-cell signaling response of NFκB under each of the five types of stimuli was measured. Different concentrations of stimuli have been added, with in total 13 different conditions (Supplementary Fig. S1). Two representative examples of the experimental data are shown: under 10 ng/ml TNF, 100 ng/ml LPS, and more examples are in Supplementary Fig. S2. In each heatmap, the color code denotes signaling activities in different cells (y-axis) over time (x-axis). Source data are provided as a Source Data file. **b** The sampled trajectories from the time-inhomogeneous Markov model (upper row) and the hidden Markov model (lower row), for the two stimulus conditions marked by the stimuli symbols. The time-inhomogeneous Markov model has 32 states, and the hidden Markov model has 64 hidden states and 32 emission states. **c** Quantification on the training performance for the two models by three measures: the ratio between KL (Kullback–Leibler) divergence (of sample and data) and entropy (of data) at each timepoint (orange), the false k-nearest neighbor probability (blue), and the rescaled log-likelihood (green). Each dot is the mean value of the measure on all the 13 stimulus conditions, and the error bar denotes the standard deviation on the 13 stimulus conditions. The numbers of hidden states and emission states have a fixed ratio 2 here. The cases of ratio 1 and individually varying the number of states can be found in Supplementary Fig. S4, while the overfitting quantification in Supplementary Fig. S5.

**Dynamical mutual information and comparison with the previous approaches**. The trained hidden Markov model allows us to infer the trajectory probability characterized by its dynamical features. We now estimate the dMI based on the summarized workflow (Fig. 1 and Supplementary Fig. S9), and compare it with the previous methods[18,27,28]. The dMI estimation has also been validated by a minimal model of a hidden Markov process (Supplementary Fig. S10).

For the NFκB signaling channel stimulated by the 13 different immune stimulus conditions (Supplementary Fig. S1), the maximum dMI rises with time (Fig. 3a), implying that distinct dynamical patterns are present at all times (Supplementary Fig. S2). Random permutation of timepoints decreases the dMI (Fig. 3a and Supplementary Fig. S12), indicating that the genuine ordering of timepoints provides information. After random permutation, the maximum dMI rises to about 1 bit as responses can be identified due to response amplitudes (Supplementary Fig. S11) but that distinct stimuli become less distinguishable.

We also investigated the dependence of dMI on key parameters. While it is not sensitive to the subsample size of trajectories following model training (Supplementary Fig. S13a), it depends on the number of hidden states (Supplementary Fig. S13b) and emission states (Supplementary Fig. S13c). Optimizing training performance while guarding against overfitting, we chose 64 hidden and 32 emission states as optimal. Using half of the trajectories (Supplementary Fig. S13d), shorter trajectories (Supplementary Fig. S13e, f) led to overfitting that inflates dMI estimates. The dMI values remained similar for bootstrapped replicates of randomly sampled data (response trajectories) with replacement (Supplementary Fig. S13g). The

difference increases over time, which may be attributed to the inaccuracy of model training at late timepoints, when signaling responses are less active and model training is more affected by measurement noise. It also indicates how limited availability of data impairs model training. Increasing the number of measured trajectories will improve the accuracy of model and dMI.

In comparison, MI calculated at single timepoints ignores trajectory information entirely (Fig. 3b). The vector method cannot consider more than around ten consecutive timepoints, because sampling the vectorial distribution becomes inaccurate, and does not distinguish the dynamical patterns when timepoints are aligned properly. Furthermore, the decoding-based method[28] that gives a lower bound on MI needs a minimal number of timepoints (Supplementary Fig. S14) and has little appreciation for additional information in longer timecourses (Fig. 3c), indicating an incomplete discrimination on the temporal dynamical patterns. We note that all the methods give MI estimates of 1–2 bits, less than the ideal limit $\log_2 13 \approx 3.7$ bits. The loss of the information can be caused by the molecular noise that governs signaling responses. Indeed, the maximum dMI are underestimates of the true biological value because all measurements are subject to technical noise or uncertainty.

**Pairwise stimuli distinction occurs in different temporal orders**. The dMI can reveal distinct temporal orders of distinguishing various stimulus pairs. To investigate the temporal order of stimulus-distinction, we took pairwise stimulus conditions from the 13 conditions in Fig. 2 and obtained the maximum dMI when using pairwise conditions (Supplementary Fig. S15). As representative examples, three distinct temporal profiles of information

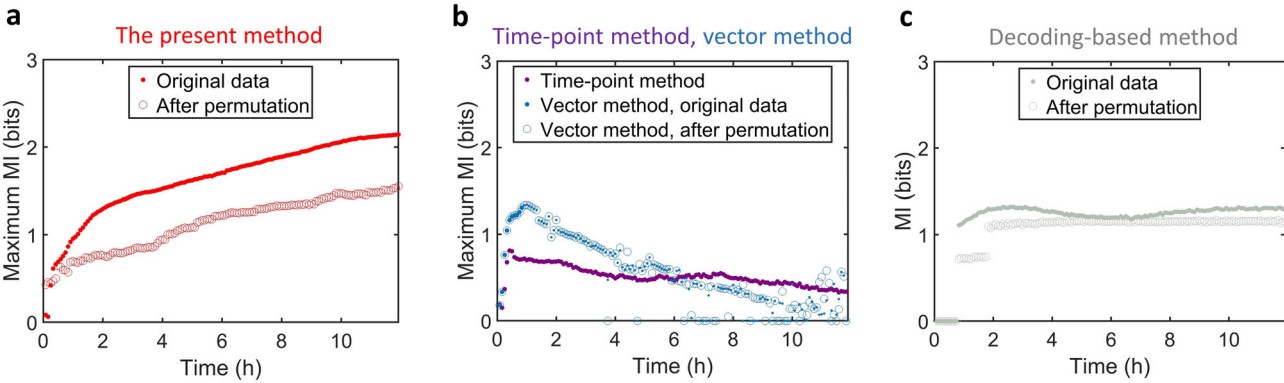

**Fig. 3 Dynamical mutual information and comparison with the previous approaches.** We used all the 13 different stimulus conditions to calculate the mutual information. **a** The maximum dMI increases with time and is diminished when timepoints are randomly shuffled. To be consistent with other MI measures, all y-axis of the maximum dMI is labeled as "Maximum MI." **b** The timepoint method[18] and the vector method[27] do not capture information in the full signaling trajectory. The vector method suggests that mutual information decreases after around 10 timepoints, while the trajectory ensembles retain substantial differences, and is independent of the ordering of timepoints. **c** The decoding-based method[28] (using the first ten principle components) provides a lower bound on mutual information (y-axis is "MI" without "Maximum") and does not reveal how mutual information accumulates over time. The random permutation of timepoints does not dramatically affect the saturated value. Source data are provided as a Source Data file.

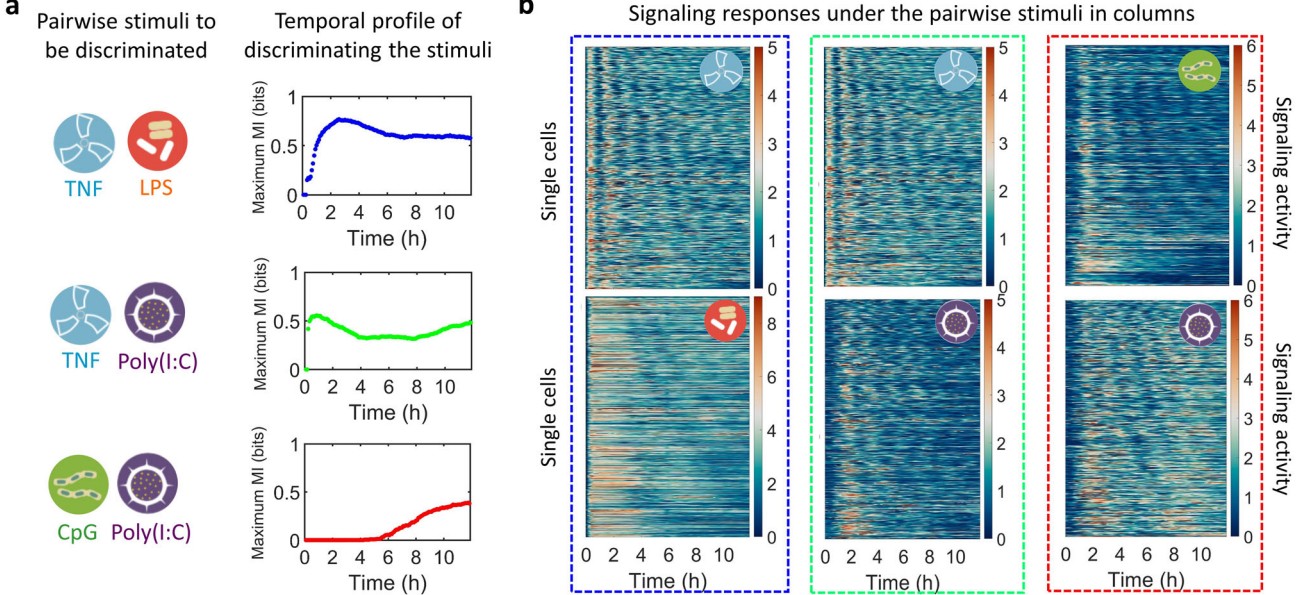

**Fig. 4 Pairwise stimuli distinction occurs in different temporal orders. a** The maximum dMI for three pairs of stimulus conditions chosen from the dataset in Fig. 2: 10 ng/ml TNF and 100 ng/ml LPS (blue); 10 ng/ml TNF and 10 μg/ml Poly(I:C) (green); 100 nM CpG and 100 ng/ml Poly(I:C) (red). **b** The heatmaps of the NFκB response for the corresponding stimuli pairs grouped in columns. The time regime with relatively distinct dynamical patterns corresponds to the high maximum dMI in (**a**). Source data are provided as a Source Data file.

transmission were revealed (Fig. 4a): (1) an early increase with sustained high MI (TNF vs LPS); (2) a rapid early increase with a drop in the intermediate phase (TNF vs Poly(I:C)); and (3) a late accumulation (Poly(I:C) vs CpG).

Examining the data directly confirmed these conclusions. First, TNF and LPS show distinct NFκB signaling dynamics through the timecourse. Second, TNF and poly(I:C) have distinct responses in the first hour, where only TNF leads to a peak of activity, and give similar responses in an intermediate phase leading to a decrease in MI. Third, CpG and poly(I:C) are different only in the late phase (Fig. 4b). Undertaking the same pairwise comparison with the previous timepoint method[18], vector method[27], and decoding-based method[28] does not provide the temporally resolved information accumulation (Supplementary Fig. S16) that the dMI provides.

To investigate the robustness of dMI calculations, we used the replicate data to replace one of the two in each pairwise conditions in Fig. 4. The dMI by using the replicates are similar (Supplementary Fig. S17), and the small differences can be attributed to the variations of the measured data between the replicates, as shown by the corresponding heatmaps. The dMI robustly reveals such differences in the replicates.

To explore how the information of the NFκB signaling can be decoded by the responsive genes, we analyzed the data of NFκB-responsive genes[41] in the same conditions (cells and stimuli) measured at 1, 3, and 8 h. From the calculated correlation between the dMI values and the absolute difference of gene expression fold change between the pairwise stimuli in Fig. 4, a large proportion of the genes track the signaling information of NFκB (Supplementary Fig. S19). We also identified representative

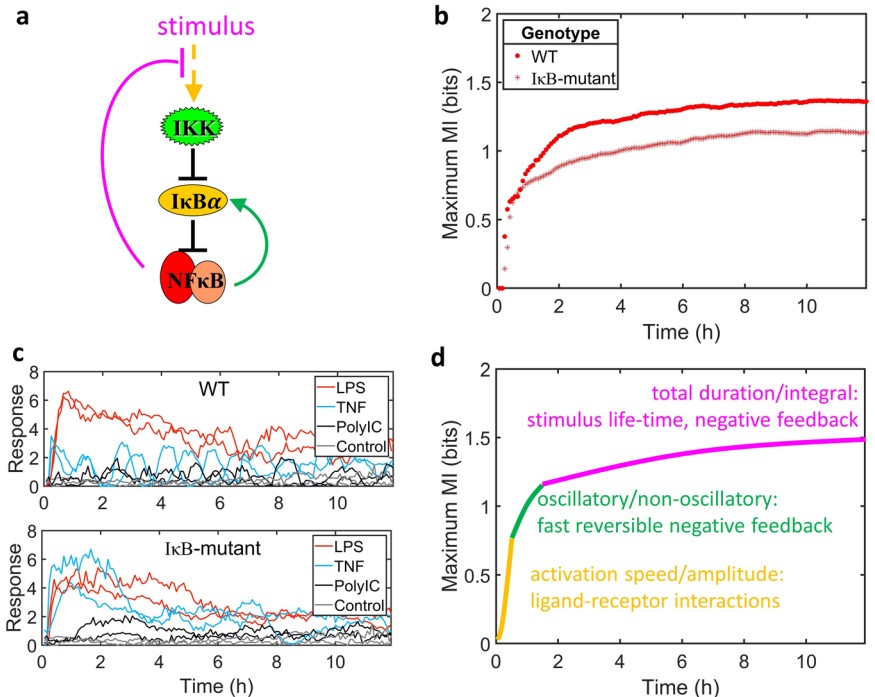

**Fig. 5 Phases of information accumulation may be mapped to regulatory motifs. a** Schematic of NFκB signaling with molecular mechanisms operating at different timescales shown in different colors. The IκB-mutant has a reduced induction of IκBα by NFκB (green arrow). To investigate the role of the feedback, we considered three stimuli (10 ng/ml TNF, 100 ng/ml LPS, 50 μg/ml Poly(I:C)) that lead to distinct signaling dynamical patterns between the genotypes. **b** The maximum dMI for NFκB responses to three stimuli (10 ng/ml TNF, 100 ng/ml LPS, 50 μg/ml Poly(I:C)) in wild-type and mutant cells lacking the IκBα feedback. The mutant shows a diminished dMI increase between 1 and 2 h. **c** Representative response trajectories for the two genotypes, illustrating that mutant cells lack oscillatory responses. **d** Schematic indicating how specific phases of information accumulation may be mapped to different molecular mechanisms that mediate specific dynamical features in the responses. The second phase was diminished in the IκB-mutant that is defective in producing oscillations; the first and third phases are conjectured to be mediated by activation amplitude and total duration of signaling, respectively. Source data are provided as a Source Data file.

genes whose expression patterns appear to follow the information accumulation. These examples suggest that the temporal ordering of discriminating immune threats may be harnessed by immune responsive target genes to sequence successive phases of immune responses. Further testing of this observation may involve knockin fluorescent reporters that provide temporal trajectory information of responsive genes at single-cell resolution and optogenetic approaches[19] that may avoid the co-activation of other factors involved in gene expression control.

**Phases of information accumulation may be mapped to regulatory motifs**. The NFκB signaling pathway consists of a series of molecular circuit motifs, including negative feedback loops[42], that operate at different timescales (Fig. 5a). One prominent feedback loop is mediated by IκBα that has the potential to provide oscillatory responses with a 1–2-h period. To test whether the information accumulation timecourse by the dMI provides insight on the timescales of regulatory mechanisms, we measured NFκB activation in response to three stimuli (10 ng/ml TNF, 100 ng/ml LPS, 50 μg/ml Poly(I:C)) in control and mutant cells deficient in IκBα-feedback (IκB-mutant). Both control and mutant cells showed a rapid increase of the dMI in the first hour, but then diverged during the second hour in which mutant cells showed a diminished increase of the dMI than control cells; after that, both cell types showed a slow but steady information gain for the duration of the 12-h timecourse (Fig. 5b).

Examining the NFκB timecourses confirmed that the second hour shows stimulus-specific deployment of an oscillatory pattern in control cells, which is abrogated in mutant cells (Fig. 5c). Thus, we could begin to map stimulus-specific dynamical features

mediated by circuit motifs to specific phases of the information accumulation timecourse (Fig. 5d). Our experimental evidence identified oscillations contributing to information accumulated in the second hour. In the first hour we speculate that stimulus-specific activation speed and amplitude provide for the first around 1 bit of information, which is largely a function of the ligand-receptor interaction and signaling adaptor activation properties. In the late phase, slow information accumulation may be due to stimulus-specific integral and duration of activity, which in turn is mediated by differential stimulus lifetimes and irreversible negative feedback loops[42]. The insights could not be fully obtained from the previous information-theoretic approaches (Supplementary Figs. S20 and S21).

We further investigated the temporal phases of dMI accumulation by generating NFκB response data with a mathematical model of the NFκB signaling network formulated by[38] and used in[43] with distributed parameters. We perturbed key parameters in the signaling network, which affect either mainly the responses' early activation, the intermediate-phase oscillation, or the sustained activity. We then calculated the dMI by considering the pairwise comparison of the data generated in the presence or absence of the signaling mechanism (Supplementary Fig. S22). Specifically, for the perturbation of the early-phase responses (modifying the ligand-receptor interaction), the dMI increases in a few minutes. When altering the intermediate-phase oscillation (reducing the negative feedback of IκBα), the dMI increases at around 1 h when the negative feedback produces an oscillatory pattern. For the perturbation on the sustained features (modifying the stimulus lifetime), the dMI increases after about 2 h, indicating the information gain derives from signaling duration

and integral. The analysis supports the hypothesis in Fig. 5d, and may prompt future experimental confirmation.

## Discussion

We have developed a quantitative framework to estimate the information accumulation encoded in the dynamics of bio-chemical signaling molecule. Specifically, we have used time-inhomogeneous Markov and hidden Markov models to learn such dynamics, which enabled the calculation of dMI to reveal the temporal profile of how information for distinguishing stimulus conditions becomes available. We demonstrated the workflow by applying it to NFκB signaling responses, where we found that some immune threats are recognized by cells more rapidly than others.

The ideal measurement of signaling responses for information-theoretic analysis would involve the same set of cells responding to different inputs, such that the potential differences between cell populations can be avoided. In practice, this is not possible as the response to one stimulus will affect the cells to the second sti-mulus. Therefore, in order to compare signaling responses of naïve cells, we split the single population of cells into physically separated compartments stimulated with a single stimulus. It is possible that the physical separation causes the subpopulations to be distinct. This is mitigated if the stimulations are done at the same time in parallel on the same microscope run, with the microscope objective gathering the data from compartment to compartment for each timepoint. To address the scenario when experiments cannot be done in parallel, we have investigated replicates produced at different times (Supplementary Fig. S17); these show only minor differences in dMI estimates.

We considered two types of stochastic models to learn sig-naling dynamics. The time-inhomogeneous Markov model is generally better at learning synchronized features, and the hidden Markov model is more suitable for data with history dependence, such as oscillatory trajectories with various frequencies. The training performance indicates that neither model reproduces the data perfectly (Fig. 2), prompting future work for more advanced machine learning models[35,36,44,45].

The proper model choice also depends on the numbers of cells and timepoints available. For the NFκB dataset with the hidden Markov model, ~500 cells with ~150 timepoints ensure the exis-tence of the proper number of parameters to achieve high model performance and avoid overfitting for the dMI calculation (Sup-plementary Fig. S5). Overfitting occurs when using half of the measured cells or every two timepoints (Supplementary Fig. S13). Yet, the time-inhomogeneous Markov model can be trained on just ~100 cells (Fig. 2 and Supplementary Figs. S6 and S7). However, to avoid overfitting, at least several fold of 500 cells are required, as indicated by the overestimates of dMI in NFκB, p53, and ERK datasets (Supplementary Fig. S8). While the time-inhomogeneous Markov model has a transition matrix to be fitted for each two consecutive timepoints, the hidden Markov model only requires two matrices in total, and thus demands fewer measured cells and timepoints.

The required numbers of cells and timepoints may vary with the complexity and heterogeneity of the dataset. We have estab-lished a computational protocol to determine those variables (Supplementary Fig. S9). If there is a range of parameters that achieves high model performance and avoids overfitting (e.g., Supplementary Fig. S5c), numbers of cells and timepoints are sufficient for the dataset. However, if the optimal number of parameters cannot be properly constrained, more cells and timepoints should be measured. In this work we used the same number of parameters across stimulus conditions as this is con-venient for computing conditional trajectory probabilities in Eq.

(7). Though different conditions are differentially prone to overfitting, the variation of overestimated MI was no more than 0.1 bit, as shown by the error bar in Supplementary Fig. S5b. Furthermore, the dMI values are comparable across datasets when the optimal number of parameters are found and used for each dataset.

The maximum dMI for a given dataset is obtained from the optimal weighting of different stimulus conditions. Cells are more likely to differentiate stimuli with high weights leading to higher maximum dMI than when weights are uniformly distributed (Supplementary Fig. S18). This plays a big role in dMI estimates for all stimulus conditions in Fig. 3 and less so for the pairwise conditions in Fig. 4, indicating that weighting optimization leads to better information transmission when discriminating many stimulus conditions.

The dMI is an information measure formulated in the trajec-tory space. A hundred years after formulating entropy for a static distribution[46], the trajectory entropy was defined along a single trajectory[37]. However, quantifying the trajectory probability remained challenging as the number of possible trajectory con-figurations increases exponentially with timepoints. By estab-lishing that stochastic models can be used to account for a large diversity of trajectories, we could arrive at an appropriate infer-ence on the trajectory entropy that allowed MI to be estimated[37]. Whereas the conventional entropy rate in the hidden Markov model[23,32] gives MI decaying to zero in the toy model (Supple-mentary Fig. S10), our choice of the trajectory entropy Eq. (1) leads to the proper 1 bit (2 bits) of the saturated MI for two (four) distinguishable trajectory ensembles.

The timecourse of information accumulation is determined by the stimulus-specific behavior of dynamic regulatory circuits. In the case of the NFκB response to distinct immune threats, we could identify three phases for threat distinction: an early phase of information based on mere signal activation; a middle phase of information gain due to the richer dynamical patterns involving selective deployment of oscillations; and a later phase of infor-mation determined by stimulus-specific values in total integral and duration. Indeed, the second phase was found to be selec-tively deficient in cells from a mutant lacking the IκBα-negative feedback circuit that provides for oscillations at the characteristic period of 1–2 h (Fig. 5c). Our results from analyzing the trajec-tories generated from a mathematical model of NFκB signaling network[38,43] further support that the temporal phases of infor-mation accumulation can come from identifiable dynamical fea-tures and molecular mechanisms that operate at different timescales, suggesting potential strategies for perturbing infor-mation accumulation by targeting specific regulatory motifs.

More broadly, quantifying information accumulation via the dMI reveals the information available to effector mechanisms that must appropriately respond in real time. Integrating the infor-mation from all earlier timepoints (after the stimulus was added) is consistent with how responsive gene expression is deter-mined[19–22]. The present framework is applicable to a variety of time series data, not only the direct measurements of cellular signaling responses, but also the temporal trajectories of gene expression when accurately inferred from single-cell RNA sequencing[47]. With such data at high temporal resolution, quantifying the information accumulation by combinatorial sig-naling pathways and the information flow between genes and gene sets are interesting future directions to be explored.

## Methods

The Methods section briefly outlines the model training, the performance eva-luation, and the calculation on the dMI. More descriptions of methods are given in Supplementary Notes 1–4.

## Model training

*Time-inhomogeneous Markov model.* Given an ensemble of the trajectories, we can extract a set of transition matrices as follows. We first binned the observed data points into discrete states. For every two consecutive timepoints, we counted the number of the data points with $y_n = l'$, $y_{n-1} = l$, where $l'$, $l$ are the states. Then, these counts represent the number of transitions from state $l$ to $l'$ at time $t = n - 1$ and are set as the $l'$th row and $l$th column element in the count matrix $c(y_n, y_{n-1})|_{t=n-1}$. The subscript for $t$ specifies the timepoint. The transition matrix is obtained by normalizing the count matrix by each column: $p(y_n|y_{n-1})|_{t=n-1} = c(y_n, y_{n-1})|_{t=n-1}/c(y_{n-1})|_{t=n-1}$ where $c(y_{n-1})|_{t=n-1} = \sum_{y_n} c(y_n, y_{n-1})|_{t=n-1}$. The trajectory probability for each measured trajectory can be obtained from these inferred transition probabilities.

*Hidden Markov model.* The Baum–Welch algorithm[48] was used to infer one transition matrix between hidden states, and one emission matrix between hidden states and emission states. It derives a maximum likelihood estimation on the parameters of the hidden Markov model given time series data. There are two numbers of states, i.e., of hidden states and emission states. In most analyses the ratio between the numbers of hidden states and emission states was fixed to 2:1 (Fig. 2). Supplementary Note 2 studies explored the case with a 1:1 ratio, and the case with individually varying the number of emission states or hidden states. Model training used a MATLAB toolbox package (https://www.mathworks.com/help/stats/hidden-markov-models-hmm.html).

## Model performance evaluation

*KL-divergence.* The KL-divergence at each timepoint is the relative entropy between the sampled trajectories and the true data distribution. The ratio of the KL-divergence and the entropy of the data distribution, termed as the "relative KL-divergence," was calculated (Supplementary Fig. S3a), and averaged along the timecourse.

*False k-nearest neighbor probability.* After mixing the sampled trajectories and data under each stimulus, for every sampled trajectory, false $k$-nearest neighbors were counted. A histogram of the number of false $k$-nearest neighbors (Supplementary Fig. S3b) was fitted to a binomial distribution to obtain the false $k$-nearest neighbor probability. The procedure was conducted for each stimulus condition, using different number of states and various number of nearest neighbor search (Supplementary Fig. S3). The false $k$-nearest neighbor probability 0.5 indicates an optimal mixing between the sampled trajectories and data.

*The log-likelihood.* We randomly picked 30 trajectories served as test dataset and the remaining as training data, enabling estimates of the log-likelihood of the trajectory probability of the test dataset based on the model of the training dataset. The trajectory probability decays inversely with the number of emission states after every time step. To counteract that, a rescaling procedure was implemented (Supplementary Note 2, II.B2). The rescaled log-likelihood is a logarithm function of the ratio between the trajectory probability and that from the null hypothesis of the hidden Markov model.

## Quantifying the dMI

We employed the hidden Markov model to demonstrate the procedure. The forward algorithm of the hidden Markov model[32] was used to estimate the probability of each trajectory in the data. For a given observed trajectory $y_{1:N}$ for the timepoint 1 to $N$, the joint trajectory probability ($1 < n < N$): $p(y_{1:n}, x_n) = \sum_{x_{n-1}} E(y_n|x_n)T(x_n|x_{n-1})p(y_{1:n-1}, x_{n-1})$, where $E$ is the emission matrix, and $T$ is the transition matrix for the hidden states $x_n$. The emission and transition matrices are inferred by the Baum–Welch algorithm for the given trajectory ensemble. The trajectory probability for an observed time series is given by summing over the $_K$ hidden states: $p(y_{1:n}) = \sum_{l=1}^{K} p(y_{1:n}, x_n = l)$.

Then, applying Eq. (1) led to the trajectory entropy. MI calculations employed the method in[27] to estimate the conditional and unconditional trajectory entropy. Specifically, For the set of $m_i$ trajectories $\{y_{1:N}^{i,j}\}_{m_i}$ under the $i$th stimulus, where $j$ is the index for a trajectory, the conditional trajectory entropy for the $i$th stimulus is estimated with the probability distribution of observing the trajectories[27]:

$$H\left(R_{1:n}^i|S=i\right) = -\sum_{j=1}^{m_i} \frac{1}{m_i} \log_2 p\left(R_{1:n}^i = y_{1:n}^{i,j}|S=i\right). \quad (4)$$

Given the probability of different stimulus $q_i = p(S = i)$, we get the total conditional entropy by summing over the conditional entropies for the $M$ stimuli:

$$H(R_{1:n}|S) = -\sum_{i=1}^{M} q_i \sum_{j=1}^{m_i} \frac{1}{m_i} \log_2 p\left(R_{1:n}^i = y_{1:n}^{i,j}|S=i\right). \quad (5)$$

The unconditional trajectory probabilities are obtained by weighting the conditional trajectory probabilities with the stimulus probability distribution:

$$p\left(R_{1:n}^i = y_{1:n}^{i,j}\right) = \sum_{k=1}^{M} q_k p\left(R_{1:n}^i = y_{1:n}^{i,j}|S=k\right). \quad (6)$$

The total unconditional trajectory entropy is given by:

$$H(R_{1:n}) = -\sum_{i=1}^{M} q_i \sum_{j=1}^{m_i} \frac{1}{m_i} \log_2 \left[\sum_{k=1}^{M} q_k p\left(R_{1:n}^i = y_{1:n}^{i,j}|S=k\right)\right]. \quad (7)$$

The dMI is calculated by Eqs. (2), (5), and (7), and its maximum is evaluated by Eq. (3). Compared with the method in[27], the present framework quantifies the information transmission encoded in dynamics of the trajectory space. More details about the calculations are in Supplementary Notes 1–3.

**Experimental data generation.** Using an mVenus-RelA endogenously-tagged mouse line[38], WT and IκB-mutant mice were produced (Supplementary Note 5, V. A). Bone marrow-derived macrophages generated from an mVenus-RelA knockin reporter mouse strain were stimulated with indicated ligands. A live cell microscopy workflow allowed measurement of nuclear NFκB levels at single-cell resolution. The measured fluorescence intensity was further normalized to image background levels, and baseline-subtracted by using an automated image analysis workflow (https://github.com/Adewunmi91/MACKtrack). It resulted in the datasets used here.

The UCLA Institutional Animal Care and Use Committee approved the protocol for animal research per guidance from the American Veterinary Medical Association.

**Reporting summary.** Further information on research design is available in the Nature Research Reporting Summary linked to this article.

## Data availability

The authors declare that the data supporting the findings of this study are available within the paper [and its supplementary information files]. Source data are provided with this paper.

## Code availability

The MATLAB code package dMI is available at GitHub (https://github.com/signalingsystemslab/dMI)[49] with a guideline on the website (https://sites.google.com/view/dmipackage). All the simulations were done with MATLAB version R2018b.

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

## Acknowledgements
We thank Diane Lefaudeux, Haripriya Vaidehi Narayanan, Supriya Sen, Katherine Sheu, and Ning Wang for valuable discussions. We also thank Sheng-hong Chen, Jeremy Purvis, and Galit Lahav for sharing the dataset of p53, and Sergi Regot and Markus Covert for sharing the dataset of ERK, p38, and JNK. The work was funded by NIH Grant R01AI127864 (to A.H.). Y.T is supported by Collaboratory fellowship at UCLA. A.A. was funded by the UCLA-Caltech MSTP: T32GM008042, Vascular Biology Training grant: T32HL69766, and a NRSA F31 fellowship: 1F31AI138450.

## Author contributions
Y.T., R.W., E.D. and A.H. designed research. Y.T. developed the theoretical workflow, with technical guidance from R.W., E.D. and F.X.-F.Y. A.A. generated experimental data. Y.T. analyzed data. Y.T. and A.H. wrote the paper, with critical input from all A.A., R.W., E.D. and F.X.-F.Y.

## Competing interests
The authors declare no competing interests.
