## [Peer Review File · Nature Communications]

Reviewers' Comments:

Reviewer #1:

Remarks to the Author:

Quantifying information in dynamic biochemical pathways is an important unresolved problem in biological signal processing and systems biology. Tang et al develop a new information theoretic measure, termed dynamic Mutual Information (dMI), to quantify information in dynamic data. This approach builds on the insight that Markov models (with or without hidden states) can be trained to learn temporal patterns in the data. Subsequently, transition probabilities generated in these model can be applied straightforwardly to calculate probabilities of dynamic trajectories, which can be then be used within the Shannon framework to quantify information. The method presented here is novel and potentially broadly applicable. The manuscript, particularly the supplemental information detailing the method, is clearly written. Some sections of the manuscript are underdeveloped, especially those pertaining to the interpretation and validation of results obtained from applying the method to experimental data. I suggesting addressing the following before the manuscript is suitable for publication.

Major Comments:

1) Interpretation of the dMI measure: One of the reasons the mutual information idea has not been straightforwardly applicable to dynamic data is the difficulty in extending the original Shannon framework, which is most clearly defined for discrete systems, to continuously varying systems. In order to derive the dMI measure described here, the authors have had to therefore make some simplifications to the original formulation. In general, these simplifications are clearly explained and seem justified for practical considerations. However, how these choices affect the interpretation of the dMI entity has not been satisfactorily explored.

Specifically, as the authors point out (Supplemental Information, section IB), trajectory probabilities diminish rapidly with increasing trajectory. To circumvent this issue, the authors discard the pre-factor term. This has the effect of rescaling the range of values of the dMI, as can be seen in Fig. S10 (b vs. c). In the context of signal processing, MI has typically been interpreted as a measure of the number of distinct signals that can be discriminated by the system. However, given that dMI varies from the MI in its range of values, it is not obvious that the same interpretation can still be applied to this new measure. The authors need to either show explicitly that this interpretation continues to hold true or explain how dMI corresponds to the discriminatory capacity of the system. One option is to extend the toy model of Fig S10 to demonstrate that dMI consistently converges to the number of bits expected, even when the number of signals is increased beyond 2.

2) Maximum dMI:

- a. The authors derive a 'maximum dMI' from the expression for dMI by maximizing over the probability distribution of signals, q_i . It will be useful to see the underlying values of q_i that maximize dMI in the data shown in Fig 3 and 4. Is there a concern here that the maximum dMI could be biased by a few specific signals?
- b. While an upper bound on the information capacity is useful, an alternative measure of the information capacity of the channel, which seems more intuitive, could be the dMI value for a uniform distribution of q_i , i.e. the dMI when every signal is equally probable. It will be helpful to know whether the authors have this measure; Is there a clear reason for using the maximum dMI, as defined, instead of this measure in order to quantify the capacity of the system?

3) Information accumulation over time:

- a. In Fig 3, the authors show that maximum dMI increases over time, and conclude that information is present at all times. How can this conclusion be reconciled with the observation that randomly permuted data in Fig 3a also shows a similar rate of increase beyond 2h?
- b. Is it possible to calculate dMI for specific time windows, i.e. $(n - \Delta t):n$, rather than cumulatively over whole trajectory, i.e. $1:n$? This might be a better way to compare that information present at

different timepoints after stimulation.

c. In the absence of ground truth regarding the amount of information in the system or how it trends over time, it is not fair to conclude (in Fig 3 legend) that 'the vector method suggests erroneously that mutual information decreases after around 10 timepoints...'

4) Experimental validation of described features: The utility of the dMI measure hinges on its ability to capture meaningful information present in the dynamics. It is therefore important that the authors validate some of the dMI-based features they describe in the NFkB data, such as the temporal ordering of information in different signals (Fig 4). For example, in the case of TNF vs. Poly (I:C) treatment or CpG vs Poly(I:C) treatment, one might expect that NFkB-responsive target genes show similar responses to the two signals during phases of low dMI and different responses during phases of high dMI. One way to measure the gene response at these phases could be to measure the fold change in target gene levels (measured using qRT-PCR, for instance) over a 1-2 h time window, at different timepoints post-treatment with TNF, Poly(I:C), or CpG.

5) Regulatory motifs mediating signal discrimination: The authors present an intriguing idea in Fig 5 – decomposing dMI into signal-specific dynamic features mediated by underlying circuit motifs. However, this section is currently rather speculative and underdeveloped. While it may be beyond the scope of the current work to experimentally validate all the ideas suggested in this section, the authors should consider doing this in the context of a mathematical model of the pathway. For example, the authors can deploy the model in Maity and Wollman (Plos Comp Bio 2020) to simulate NFkB responses to different signals. Biochemical parameters and regulatory interactions in the model can then be systematically altered to generate new data corresponding to perturbations of the system. Through this, should be possible to map how different regulatory motifs and biochemical parameters in the NFkB pathway could contribute to levels and trends in dMI.

Minor Comments:

1) It will be helpful to include a brief description of the reporter cell line and experimental setup used to generate the data in the main text and in Figure 2a schematics.

2) The analysis of p53 and ERK dynamics are not mentioned in the text until the discussion section, which alludes to 'advantages for specific datasets, such as p53 ...'. It will be helpful for readers if these comparisons are mentioned earlier in the main text, perhaps at the end of the section relating to Fig 2.

3) Since model training and dMI calculation are sensitive to the number and length of trajectories, it seems that it would not be possible to compare dMI across datasets if they differ in these parameters. If so, is there a way to generate a more standardized measure that can be used to directly compare different datasets?

4) Some clarifications are required in Fig 4 and corresponding section in the text. As I understand, a subset of data from Fig 2 is re-analyzed to generate the new results. This should be made clear in the text and legend. The light/dark blue regions in Fig 4b are confusing, especially since the data colormap includes blues; it is also not clear how these regions were delineated.

Reviewer #2:

None

Reviewer #3:

Remarks to the Author:

In 'Quantifying information accumulation encoded in the dynamics of biochemical signaling' Tang et al. develop a framework to determine what they call the 'dynamical mutual information' (dMI), or the mutual information encoded in temporal patterns of signaling molecules. To do this they use either a time-inhomogeneous Markov model or a hidden Markov model and find the optimal number of parameters to use. They then test their workflow using NF-kB single cell dynamic data and find that it

can quantify the capacity of cells to distinguish immune stimuli.

The application of information theory to biological signaling systems has expanded in recent years due to the availability of high-quality single-cell signaling data. However, these information theoretic approaches have either been limited to single timepoint measurements or have used time series data without accounting for the temporal order of the data. Therefore, although these approaches are calculating MI none of them can fully incorporate that cells use dynamic signal patterns to encode information about a stimulus. The authors nicely review previous efforts in the field and highlight their limitations. Throughout the paper the authors benchmark their approach against others, and this supports their conclusions that their framework is best suited to tackle the challenge of quantifying the mutual information in dynamic data, especially those with oscillations, and to determine how this accumulates over time.

The authors should be commended for their extensive supplemental data and their effort to make a website detailing the approach prior to publication. This will greatly increase the accessibility of their model to the broader single-cell signaling community. Overall, this paper represents an exciting new step in applying information theoretic approaches to understand how cells accumulate information in dynamic signaling patterns. Therefore, I would recommend this manuscript for publication as long as the following points are addressed:

- 1) The authors claim that the method is generally applicable. Although they do test their model on both p53 and MAPK single-cell data and discuss this in the supplemental data I would recommend that the authors should move some of this information/figures to the main text. This would solidify their claim that their approach is generalizable to dynamical signaling systems other than NF-kB. In addition, although these datasets are published it would be helpful to add labels so that the reader knows what stimuli were used and when without having to refer back to the original papers.
- 2) The additional signaling datasets raise two other important issues – how many cells and how many timepoints are needed for this approach? Again, the authors do discuss this in the supplement, but there should be an expanded discussion of this in the main text as these are both key decisions a user would need to make to implement the dMI approach described in this paper. It was also unclear how many cells are needed for the time-inhomogeneous Markov model – I may have missed this in the supplement but if not, it should be included. Finally, are the authors able to estimate whether the number of cells needed changes based on how heterogeneous a dataset is in signaling dynamics?
- 3) Can the authors comment on whether their choice of 64 hidden and 32 emission states as optimal for the HMM is specific to the NFkB dataset or do they anticipate that this is true for other oscillating signaling systems (p53, ERK) or non-oscillating signaling systems (p38)?

Minor points:

Lines 204-205: The authors should add to the text what their conclusion is after exploring how varying the number of hidden or emission states altered HMM performance.

Reviewer #4:

Remarks to the Author:

This paper proposes a methodology for quantification of mutual information of time course data. It is interesting, novel and potentially important.

I have one question to be resolved before publication, though.

To what extent does this definition of the mutual information depend the time resolution of the data, and on uncertainties in the data?

Reviewer #5:

Remarks to the Author:

In this manuscript, Tang and co-authors worked on a key question in cell signaling – how do cells encode extracellular inputs as temporal regulator dynamics. To address this question, one would need to quantify the amount of information encoded in the temporal dynamics, and determine how well cells can utilize such encoded information to differentiate different inputs. However, it has been challenging to accurately quantify the dynamics-encoded information. While several methods have been developed to calculate such information, they have not yet fully leveraged the time sequence of the temporal data. In this work, the authors developed a hidden Markov model-based approach to estimate the information encoded in the temporal dynamics, which performed much better compared to existing methods. This novel approach allowed the authors to decipher the temporal changes in the encoded information, and investigate different molecular mechanisms affecting information encoding. Overall, this manuscript is well written and the results are generally nicely presented. What they have revealed provide new insights into how cells encode inputs into temporal signals, and their method should be broadly applicable for understanding dynamic signal processing in different organisms. However, I have several concerns that the authors may wish to address before publication.

Major concerns:

1. A general remark on the data and the MI value. While it is a common practice in the field to quantify signal processing and information encoding with data from multiple different populations of cells, it appears that it should be better to collect the response dynamics from the same population of cells. In other words, the biological definition of “the ability to differentiate different inputs” refers to the ability of a cell to tell one input apart from the other. That said, it makes a lot more sense to calculate MI based on paired datasets, i.e., the dynamic responses of the same set of cells to different inputs. Although there are technical challenges in doing so, working on such paired datasets seems to be more biologically meaningful. It would be great if the authors can elaborate on this point in the manuscript. This point is likely related to the generally low MI values reported in the manuscript. For example, in Fig. 3a, the maximum MI is ~ 2 (i.e., corresponds to the ability to differentiate 4 inputs). And for Fig. 4a, the maximum MI is less than 1 (i.e., can't differentiate two inputs). Alternatively, it seems that the MI computed is a “technical” MI rather than the “biological” MI – i.e., cells must have done better than this.
2. About model training. It is unclear whether the authors used data from all 13 samples to train one model or they used one sample set for training each model (i.e., a total of 13 models). In line 133, the authors said that they train a Markov or hidden Markov model for each stimulus condition, what if training one model for all conditions or for each stimulus type since we can assume that under some conditions the underlying biological networks are the same. This may also reduce the effect of overfitting.
3. About model validation. It appears that the authors used in-sample data for cross-validation. I wonder if there is a way to perform out-of-sample cross-validation with data from biological replicates (that are collected on different days/experiments).
4. In fig. S7, it is somewhat strange and hard to understand that the HMM model is worse than the time-homogeneous Markov model. Although the authors provided potential explanations, I think the key issue is related to the nature of the data. The data in Fig. S7a are composed of equal segments of pre-stress and post-stress activity data, which is in sharp contrast to their own NFkB datasets in Fig. 2a and Fig. S2a, which are composed of primarily post-stress activity data. I think this is a key issue because these two types of temporal data are fundamentally different, which likely affects model fitting.
5. I am curious about the performance of dMI under different sampling frequencies (i.e. different length of interval between time points), is there a minimum time interval for accurate computation of

dMI and how is it related to the time scale of the underlying biological process?

6. Line 211-215, the authors chose the best state number for all conditions while under different conditions the underlying biological networks differ. What if choosing the best state number for each condition (even though this may contribute to overfitting)? And in line 241-242, I was confused about the relationship between overfitting and trajectory number. If reducing trajectory number can lead to overfitting, then state number cannot be set the same since the trajectory numbers of different conditions are different.

7. Biological insights. The authors said that stimuli distinction occurs in different temporal orders. I wonder if these temporal orders are related to signaling cascades in cells. Can the authors combine their results with related signaling pathways to further prove their methods and reveal more biological insights? Is there any difference for temporal profiles of dMI between within-stress-type comparison and across-stress-type comparison? If so, is it related to underlying signaling networks? And in Line 294, can the authors use the 13-sample data to prove these hypotheses?

Minor points:

1. Need to proofread the manuscript more carefully, e.g., Line 41 etc
2. Line 355, is the normalization term wrong (if normalized by rows)? What's more, since the meaning of p has changed after normalization, this symbol should be changed after normalization.
3. In the legend of Fig. S16a, "Fig. 5's panel (c)" should be "Fig. 5's panel (b)".

Tang et al "Quantifying information accumulation encoded in the dynamics of biochemical signaling" - Responses to Reviewers

We thank all the reviewers for their careful reading and inspiring comments, which have led to an improved version of the manuscript. All four reviewers agreed that the results are novel, broadly applicable, and important: "The method presented here is novel and potentially broadly applicable. The manuscript, particularly the supplemental information detailing the method, is clearly written" (Reviewer #1); "Overall, this paper represents an exciting new step in applying information theoretic approaches..." (Reviewer #2); "It is interesting, novel and potentially important" (Reviewer #3); "Overall, this manuscript is well written and the results are generally nicely presented. What they have revealed provide new insights..., and their method should be broadly applicable..." (Reviewer #4).

The reviewers made suggestions to further develop some Results sections and elaborate the Discussion. The major points include: (1) the relation of the dMI formulation to standard MI; (2) the connection of dMI to downstream responsive gene expression; (3) the validation on the hypothesis in Fig.5 by using a mathematical model of the NF κ B signaling network; (4) the choice of the proper model; (5) the criteria for the required numbers of cells and timepoints; (6) the proper number of parameters for different datasets. Accordingly, we have conducted more analyses to address these questions, provided detailed discussions, and revised presentations. An overview of the changes is shown in the following paragraph, followed by detailed responses to each of the points raised by the reviewers.

Summary of changes

Changes in the main text:

- Figures:
 1. Fig.2: added the reporter cell line in the schematic panel.
 2. Fig.4: removed the dark view on the heatmaps for better visualization.
 3. Fig.5: revised the time scale on the interpretation of the three phases of dMI accumulation, based on the results from using a mathematical model of the NF κ B signaling network.
- Results section:
 1. In the Results section for Fig.2, we included the results from applying the framework to p53 and ERK datasets.
 2. In the Results section for Fig.3, we analyzed the dMI for the randomly permuted data and discussed mutual information estimated using various methods.
 3. In the Results section for Fig.4, we mentioned that it used the data in Fig.2, and added the result of dMI when using replicates for the pairwise conditions. We also analyzed the correlation between the dMI and the difference of the NF κ B-responsive gene expressions under the pairwise conditions, and

- provided candidate gene that may track the information accumulation of NFκB signaling.
4. In the Results section for Fig.5, we added dMI calculations using simulations from a mathematical model of the NFκB signaling network, to validate the hypothesis about mapping the temporal phase of information to signaling network mechanisms.
- Discussion section:
 1. We discussed the sample-by-sample setup of measurement rather than adding subsequent stimuli to one sample.
 2. We discussed the proper model choice and its dependence on the numbers of cells and timepoints available.
 3. We provided a guidance for the required numbers of cells and timepoints, which may vary with the heterogeneity of the dataset.
 4. We clarified our choice of using the same number of parameters across stimulus conditions.
 5. We compared the dMI by using the optimal or uniform distribution of stimulus conditions.
 6. We added the result of the toy model by using multiple sets of trajectories.
 7. We discussed the dMI results using the trajectories generated from a mathematical model of the NFκB signaling network.
 - Methods section:
 1. We revised the notation on the count matrix for the time-inhomogeneous Markov model.
 2. We provided more details on the experimental steps.
 - Reference section's major changes:
 1. Added [Maity, A. & Wollman, R. PLOS Computational Biology 16, e1008011 (2020)] which also used the mathematical model of the NFκB signaling network published by [Taylor, B. et al. bioRxiv 2020.05.23.112862 (2020)].
 2. Added [Cheng, C. S. et al. Cell Systems 4, 330-343.e5 (2017)] for using their measured data of NFκB-responsive gene expressions.
 3. Changed [Chen, S. Y. et al. Cell Systems 11, 336-353.e24 (2020)] from bioRxiv to its published version.

Changes in Supplemental Information:

- Figures:
 1. Fig.S7: for the ERK dataset, added the trajectories generated from hidden Markov models trained by two time windows before and after the stimulus.
 2. Fig.S10: added the maximum dMI for 4 different sets of trajectories, by using the two entropy formulas with or without the pre-factor.
 3. Fig.S11: new figure with the heatmaps of the trajectory ensembles after random permutation on timepoints.
 4. Fig.S13: added panels for the dMI for the time window from 2 hours to 12 hours and the dMI for the data by using every two time points.
 5. Fig.S17: new figure of the dMI for the pairwise stimulus conditions with using replicate datasets.

6. Fig.S18: new figure about the optimal stimulus distribution q_i for Fig.3 and Fig.4, and the dMI when using uniform stimuli distribution.
 7. Fig.S19: new figure about the correlation between NF κ B-responsive gene expression trajectories and the dMI.
 8. Fig.S22: new figure for the dMI by using a mathematical model of the NF κ B signaling network.
- Tables:
 1. Table.S3: added the experimental date for each NF κ B dataset.
 - Supplementary II.B.3: added discussions about the maximum dMI after the random permutation of timepoints, the dependence of dMI on the stimulus probability distribution, and the dMI for the pairwise stimulus conditions with using replicate datasets.
 - Supplementary II.B.4: this new subsection discusses in more detail about the proper number of states in the model to account for the available datasets, the dMI versus the number of timepoints and time window, and the dMI versus the number of measured cells.
 - Supplementary II.D.3: analyzed the results of the toy model when using multiple sets of time series.
 - Supplementary III.A.3: demonstrated the application of hidden Markov model to the ERK dataset for two time windows.
 - Supplementary IV.D.1: this new subsection describes how dMI accumulation relates to the expression of selected NF κ B-responsive genes, and the dMI by using the trajectories generated from a mathematical model of the NF κ B signaling network.

Detailed reply to Reviewer #1

Original reports are typed in blue color and our response in black. Please note the numbers of equations and figures refer to the revised manuscript, and the numbers of references refer to the list after the reply to each reviewer.

Quantifying information in dynamic biochemical pathways is an important unresolved problem in biological signal processing and systems biology. Tang et al develop a new information theoretic measure, termed dynamic Mutual Information (dMI), to quantify information in dynamic data. This approach builds on the insight that Markov models (with or without hidden states) can be trained to learn temporal patterns in the data. Subsequently, transition probabilities generated in these model can be applied straightforwardly to calculate probabilities of dynamic trajectories, which can be then be used within the Shannon framework to quantify information. The method presented here is novel and potentially broadly applicable. The manuscript, particularly the supplemental information detailing the method, is clearly written.

We thank the reviewer for the careful reading of our manuscript. We are also grateful to the reviewer's pithy summary of the major results and the assessment that the results novel, broadly applicable, and clearly presented.

Some sections of the manuscript are underdeveloped, especially those pertaining to the interpretation and validation of results obtained from applying the method to experimental data. I suggesting addressing the following before the manuscript is suitable for publication.

We appreciate the reviewer's comments. Below, we have done our best to address all the comments, and further developed the manuscript by including additional results and analyses.

Major Comments:

1) Interpretation of the dMI measure: One of the reasons the mutual information idea has not been straightforwardly applicable to dynamic data is the difficulty in extending the original Shannon framework, which is most clearly defined for discrete systems, to continuously varying systems. In order to derive the dMI measure described here, the authors have had to therefore make some simplifications to the original formulation. In general, these simplifications are clearly explained and seem justified for practical considerations. However, how these choices affect the interpretation of the dMI entity has not been satisfactorily explored.

We agree with the reviewer on that Shannon's original framework was mainly defined for discrete probability. To extend it to the continuous probability space, we started from the fundamental concept of differential entropy Eq.(S1.3). It has been established that the differential entropy is a proper extension for the given problem in a continuous state space ¹, which provides a solid starting point to formulate MI in trajectory space. To

calculate the differential entropy given finitely measured data does require careful implementation – describing the implementation is a key aspect of the manuscript.

Specifically, as the authors point out (Supplemental Information, section IB), trajectory probabilities diminish rapidly with increasing trajectory. To circumvent this issue, the authors discard the pre-factor term. This has the effect of rescaling the range of values of the dMI, as can be seen in Fig. S10 (b vs. c). In the context of signal processing, MI has typically been interpreted as a measure of the number of distinct signals that can be discriminated by the system. However, given that dMI varies from the MI in its range of values, it is not obvious that the same interpretation can still be applied to this new measure. The authors need to either show explicitly that this interpretation continues to hold true or explain how dMI corresponds to the discriminatory capacity of the system. One option is to extend the toy model of Fig S10 to demonstrate that dMI consistently converges to the number of bits expected, even when the number of signals is increased beyond 2.

We first would like to mention that the dMI values in Fig.S10b,c are beyond rescaling alone: They have distinct temporal profiles. Specifically, in Fig.S10b the dMI keeps increasing, which reveals the accumulation of mutual information to distinguish the two distinct sets of time series. However, in Fig.S10c the dMI increases a little above 0 and soon erroneously converges to zero for the two distinguishable sets of time series.

As noticed by the reviewer, we have discarded the pre-factor in the standard $p \cdot \log(p)$ entropy formula, and used the trajectory entropy $\log(p)$ as the entropy for each single trajectory². We then take the average $\log(p)$ for the measured trajectories under each stimulus as formulated in¹. This treatment constitutes a Monte-Carlo type of sampling in the trajectory probability space and calculated the mean entropy of the measured trajectories. Once the measured trajectories are sufficient to cover the major configurations of the full probability distribution, the average trajectory entropy approximates to the entropy of the full distribution in the trajectory space. Then, the dMI with such treatment has the same interpretation of quantifying the number of distinct stimulus conditions.

Following the reviewer's good suggestion, we now have considered multiple (4 different) stimulus conditions ("signals") in the toy model, and estimated the dMI as a measure to differentiate these conditions. We used 4 sets of trajectories generated from the toy hidden Markov model, which are largely distinct by the transition and emission probabilities (for one set of trajectories the transition rate p varies as specified by color in the figure). In Fig.S10, also inserted below, the trajectory entropy formula leads to the expected 2 bits of dMI in saturation, when the 4 different sets of trajectories can be fully discriminated over time. The entropy rate formula $(1/n) \cdot \log(p)$ still gives zero mutual information. Overall, these results demonstrate that the dMI can measure the number of distinct signals in the same way as the standard MI formulation.

For 4 different sets of trajectories

2) Maximum dMI:

a. The authors derive a ‘maximum dMI’ from the expression for dMI by maximizing over the probability distribution of signals, q_i . It will be useful to see the underlying values of q_i that maximize dMI in the data shown in Fig 3 and 4. Is there a concern here that the maximum dMI could be biased by a few specific signals?

We now provide the optimal distribution q_i for Fig.3 and Fig.4 in Fig.S18, as inserted below. The results show that the maximum dMI depends on the distribution of signals, as it should. The maximization procedure gives the optimal distribution that can bias to certain stimulus conditions (“signals”). Indeed, the bias is informative for the biological process, as it tells that certain stimulus conditions are more distinguishable by cells than others. Cells are more likely to differentiate conditions with high distributed values.

b. While an upper bound on the information capacity is useful, an alternative measure of the information capacity of the channel, which seems more intuitive, could be the dMI value for a uniform distribution of q_i , i.e. the dMI when every signal is equally probable. It will be helpful to know whether the authors have this measure; Is there a clear reason for using the maximum dMI, as defined, instead of this measure in order to quantify the capacity of the system?

As a comparison, we now also use uniform distribution of q_i to calculate the dMI for both Fig.3 and Fig.4 (Fig.S18) as shown above. The optimal distribution leads to higher maximum dMI than the uniform distribution. For all stimulus conditions in Fig.3, the maximum dMI is much higher than that from uniform stimuli distribution, indicating that the maximization leads to better information transmission when discriminating many stimulus conditions. For the pairwise condition, the maximum dMI is only slightly higher, because the room to optimize the distribution of two stimuli is limited.

The maximum dMI is used to quantify the maximum ability of the cells to discriminate different signals, which is the central goal of calculating the mutual information for stimuli-discrimination^{1,3}. When the signals of various types and concentrations are exhaustively sampled by experiments, the maximum dMI approaches the true channel capacity, as a measure of the maximum capacity for information transmission by the signaling channel. Here, we focus on the maximum dMI, providing an upper limit of information that cells can acquire from the available datasets/stimuli.

3) Information accumulation over time:

a. In Fig 3, the authors show that maximum dMI increases over time, and conclude that information is present at all times. How can this conclusion be reconciled with the observation that randomly permuted data in Fig 3a also shows a similar rate of increase beyond 2h?

The maximum dMI of random permutation increases over time beyond 2 hours, because the trajectory ensembles under various stimulus conditions can still be discriminated. For example, as shown by the heatmaps inserted below, the trajectory ensembles after random permutation have differences even after 2 hours. There are trajectory ensembles with many high (over 5), middle (around 4) and low (below 3) response amplitudes. Such differences are revealed by the maximum dMI of permuted data that continues to rise after 2 hours.

b. Is it possible to calculate dMI for specific time windows, i.e. $(n - \Delta t):n$, rather than cumulatively over whole trajectory, i.e. $1:n$? This might be a better way to compare that information present at different timepoints after stimulation.

We would like to mention that the signaling response happens soon after the stimulus is added. Thus, we start from the time point after the stimulus was added, such that the dMI can quantify stimuli discrimination in an accumulative way. This type of information is particularly relevant to biological processes where the accumulated signaling activities can determine gene expression⁴⁻⁷.

Below we also calculated the dMI for a time window from 2 hours to 12 hours (by using this time window, most data is kept, such that the overfitting does not dramatically increase). We repeat the computational protocol of model training and dMI calculation by using the trajectories only in this time window. As shown in the right panel here, the dMI has nonzero value at the beginning of 2 hours, which is lower than the result at 2 hours from the full time window (left panel). The reason is that the earlier time points provide information cumulatively for the full time window case. Note that the dMI from the shorter time window may be an overestimate due to the overfitting (Fig.S13e). Overall, the dMI from late time window quantifies the extent of discriminating stimuli in that time window, but does not reflect how cells discriminate stimuli cumulatively after sensing the stimuli.

c. In the absence of ground truth regarding the amount of information in the system or how it trends over time, it is not fair to conclude (in Fig 3 legend) that ‘the vector method suggests erroneously that mutual information decreases after around 10 timepoints...’.

We now revise the text to be more precise, as “the vector method suggests that mutual information decreases after around 10 time points while the trajectory ensembles retain substantial differences”. By design, the vector method was restricted to low numbers of timepoints: sampling the vectorial distribution from limited data becomes inaccurate as the configurations of the vector space increases exponentially with respect to the number of its dimension. Therefore, the vector method has seldom been used when the number of timepoints is over around 10. The subsequent methods tend to overcome such difficulty, including the decoding-method⁸ that gives MI estimates without a dramatic decrease in the later time window (Fig.3). The current method can also overcome this difficulty, and is further able to extract the information encoded in dynamical patterns.

4) Experimental validation of described features: The utility of the dMI measure hinges on its ability to capture meaningful information present in the dynamics. It is therefore important that the authors validate some of the dMI-based features they describe in the NFkB data, such as the temporal ordering of information in different signals (Fig 4). For example, in the case of TNF vs. Poly (I:C) treatment or CpG vs Poly(I:C) treatment, one might expect that NFkB-responsive target genes show similar responses to the two signals during phases of low dMI and different responses during phases of high dMI. One way to measure the gene response at these phases could be to measure the fold change in target gene levels (measured using qRT-PCR, for instance) over a 1-2 h time window, at different timepoints post-treatment with TNF, Poly(I:C), or CpG.

We thank the reviewer for the great suggestion. As a pilot study, we now analyze the data of NFkB-responsive genes in ⁹ by our lab in the same conditions (cells and stimuli). We used the measured NFkB-responsive gene expression (clusters B and C in Fig.5 of ⁹) at three time points (1, 3, 8 hours), under the treatment of TNF, Poly(I:C), or CpG separately. We first calculated the correlation between the dMI values and the absolute difference of gene expression fold change between the pairwise stimuli, at the time points 0, 1, 3, 8 hours. From the histogram of this correlation (the panel b below), a large proportion of genes track the signaling information of NFkB. More time points of expression measurement may be required to further validate the correlation.

We then chose representative genes whose expression patterns appear to follow the information accumulation. That is, the large absolute differences of gene expression fold change between the pairwise stimuli correlate with high dMI values, and vice versa. For example, the protease inhibitor *Serpina3f* ¹⁰ is induced when macrophages sense bacteria (LPS) locking up infected tissues to prevent bacterial spread, but not when they sense TNF which derives from neighboring cells. The distinction is sensed early and is sustained, which matches with the dMI between LPS and TNF. See more descriptions on the genes and discussions in SI Section IV.D.1.

While these observations are promising, we would like to point out the following limitations and challenges when relating gene expression to signaling responses. First, the mutual information quantifies the discrimination among stimuli based on the single-cell data, whereas the gene expression data is based on a population-level bulk measurement. Future studies may employ knockin fluorescent reporters to report single-cell gene expression dynamics following stimulation. Second, while these genes were selected to be NFkB target genes, we cannot exclude the involvement of other factors that may be activated by the stimuli and may play a role in gene expression control, thus confounding the analysis. Future studies may address this limitation by using experimental approaches such as optogenetic control of transcription factors⁴ that avoid co-activating other pathways. Nevertheless, the data presented here can be a motivation for further exploration of this topic.

5) Regulatory motifs mediating signal discrimination: The authors present an intriguing idea in Fig 5 – decomposing dMI into signal-specific dynamic features mediated by underlying circuit motifs. However, this section is currently rather speculative and underdeveloped. While it may be beyond the scope of the current work to experimentally validate all the ideas suggested in this section, the authors should consider doing this in the context of a mathematical model of the pathway. For example, the authors can deploy the model in Maity and Wollman (Plos Comp Bio 2020) to simulate NFkB responses to different signals. Biochemical parameters and regulatory interactions in the model can then be systematically altered to generate new data corresponding to perturbations of the system. Through this, should be possible to map how different regulatory motifs and biochemical parameters in the NFkB pathway could contribute to levels and trends in dMI.

Following the reviewer's insightful suggestion, we have used the mathematical model formulated by¹¹ and used in¹² with distributed parameter values, to generate data for dMI calculations, as shown in the figure below. To validate the hypothesis in Fig.5, we perturbed key parameters of the signaling network, which affect either mainly the responses' early activation, the intermediate-phase oscillation, or the sustained activity. We then calculated the dMI by considering the pairwise comparison of data generated in the presence or absence of the signaling mechanism. Specifically, for the perturbation on the early-phase responses' activation by modifying the interaction of LPS and receptor (heatmaps 1,2), the dMI increases in a few minutes. When perturbing the intermediate-phase oscillation by knocking down the negative feedback of IκBa (heatmaps 3,4), the dMI increases around 1 hour when the negative feedback produces an oscillatory pattern. For the perturbation on the sustained features by changing the stimulus life-time (heatmaps 1,5), the dMI increases relatively late, indicating the gain of information from the sustained features of duration and integral.

We note that there is no clear-cut separation of the features. For example, the dMI for heatmaps 1,2 may also come from late sustained features, and the dMI for heatmaps 1,5 can be attributed to both oscillation and sustained features, such that the level and trends in dMI coming from a single feature may not be separated. Still, the analysis reveals that the temporal phases of information accumulation can come from features in

different time scales, and helps demonstrate potential strategies to perturb the information accumulation by specific regulatory motifs. These findings as hypothesized in Fig.5d warrant future experimental confirmation.

Minor Comments:

1) It will be helpful to include a brief description of the reporter cell line and experimental setup used to generate the data in the main text and in Figure 2a schematics.

We added more detailed descriptions on the reporter cells, which are primary bone-marrow-derived macrophages (not a cell line), and experimental steps in the section “Experimental data generation” of the main text. We also put the reporter cell, “Bone marrow-derived macrophages generated from an mVenus-RelA knockin reporter mouse strain”, in Fig.2a.

2) The analysis of p53 and ERK dynamics are not mentioned in the text until the discussion section, which alludes to ‘advantages for specific datasets, such as p53 ...’.

It will be helpful for readers if these comparisons are mentioned earlier in the main text, perhaps at the end of the section relating to Fig 2.

We thank the reviewer for this suggestion, and now have discussed p53 and ERK datasets in the Results section of Fig.2, when we describe the model training. It helps to compare the performance of the stochastic models for different datasets and support the broad applicability of the method.

3) Since model training and dMI calculation are sensitive to the number and length of trajectories, it seems that it would not be possible to compare dMI across datasets if they differ in these parameters. If so, is there a way to generate a more standardized measure that can be used to directly compare different datasets?

We would like to mention that the dMI values are comparable across datasets when the optimal number of parameters are found and used for each dataset. This optimal number of parameters may differ across datasets, and one needs to search for the optimal number, which should give good model performance and avoid overfitting (Fig.S5). When such optimal range is found, the dMI (up to a small range of variation due to the possible variation on the proper range of parameters' number) is an exact value of the information in bits, which may be compared among datasets. The dMI value is also comparable to the results from the other methods (Fig.3) in terms of their absolute values. For example, the ranges of the maximum MI are all around 1 bit at 1 hour for these methods.

4) Some clarifications are required in Fig 4 and corresponding section in the text. As I understand, a subset of data from Fig 2 is re-analyzed to generate the new results. This should be made clear in the text and legend. The light/dark blue regions in Fig 4b are confusing, especially since the data colormap includes blues; it is also not clear how these regions were delineated.

We now have explicitly specified in text and Fig.4's legend that Fig.4 has re-analyzed the dataset of Fig.2. It used the data in a different way with a focus on discriminating the pairwise stimuli. We have also removed the light/dark blue regions in Fig.4b to avoid the confusion. Without the light/dark region, it is still observable that the time regime with relatively distinct dynamical patterns corresponds to the high maximum dMI.

References:

1. Selimkhanov, J. *et al.* Accurate information transmission through dynamic biochemical signaling networks. *Science* **346**, 1370–1373 (2014).
2. Seifert, U. Entropy Production along a Stochastic Trajectory and an Integral Fluctuation Theorem. *Phys. Rev. Lett.* **95**, 040602 (2005).
3. Cheong, R., Rhee, A., Wang, C. J., Nemenman, I. & Levchenko, A. Information Transduction Capacity of Noisy Biochemical Signaling Networks. *Science* **334**, 354–358 (2011).

4. Chen, S. Y. *et al.* Optogenetic Control Reveals Differential Promoter Interpretation of Transcription Factor Nuclear Translocation Dynamics. *Cell Systems* **11**, 336-353.e24 (2020).
5. Hao, N. & O'Shea, E. K. Signal-dependent dynamics of transcription factor translocation controls gene expression. *Nature Structural & Molecular Biology* **19**, 31–39 (2012).
6. Purvis, J. E. & Lahav, G. Encoding and Decoding Cellular Information through Signaling Dynamics. *Cell* **152**, 945–956 (2013).
7. Sen, S., Cheng, Z., Sheu, K. M., Chen, Y. H. & Hoffmann, A. Gene Regulatory Strategies that Decode the Duration of NFκB Dynamics Contribute to LPS- versus TNF-Specific Gene Expression. *Cell Systems* **10**, 169-182.e5 (2020).
8. Granados, A. A. *et al.* Distributed and dynamic intracellular organization of extracellular information. *PNAS* **115**, 6088–6093 (2018).
9. Cheng, C. S. *et al.* Iterative Modeling Reveals Evidence of Sequential Transcriptional Control Mechanisms. *Cell Systems* **4**, 330-343.e5 (2017).
10. Heit, C. *et al.* Update of the human and mouse SERPINE gene superfamily. *Human Genomics* **7**, 22 (2013).
11. Taylor, B., Adelaja, A., Liu, Y., Luecke, S. & Hoffmann, A. Identification and physiological significance of temporal NFκB signaling codewords deployed by macrophages to classify immune threats. *bioRxiv* 2020.05.23.112862 (2020).
12. Maity, A. & Wollman, R. Information transmission from NFκB signaling dynamics to gene expression. *PLOS Computational Biology* **16**, e1008011 (2020).

Detailed reply to Reviewer #2

Original reports are typed in blue color and our response in black. Please note the numbers of equations and figures refer to the revised manuscript.

In 'Quantifying information accumulation encoded in the dynamics of biochemical signaling' Tang et al. develop a framework to determine what they call the 'dynamical mutual information' (dMI), or the mutual information encoded in temporal patterns of signaling molecules. To do this they use either a time-inhomogeneous Markov model or a hidden Markov model and find the optimal number of parameters to use. They then test their workflow using NF- κ B single cell dynamic data and find that it can quantify the capacity of cells to distinguish immune stimuli.

The application of information theory to biological signaling systems has expanded in recent years due to the availability of high-quality single-cell signaling data. However, these information theoretic approaches have either been limited to single timepoint measurements or have used time series data without accounting for the temporal order of the data. Therefore, although these approaches are calculating MI none of them can fully incorporate that cells use dynamic signal patterns to encode information about a stimulus. The authors nicely review previous efforts in the field and highlight their limitations. Throughout the paper the authors benchmark their approach against others, and this supports their conclusions that their framework is best suited to tackle the challenge of quantifying the mutual information in dynamic data, especially those with oscillations, and to determine how this accumulates over time.

The authors should be commended for their extensive supplemental data and their effort to make a website detailing the approach prior to publication. This will greatly increase the accessibility of their model to the broader single-cell signaling community. Overall, this paper represents an exciting new step in applying information theoretic approaches to understand how cells accumulate information in dynamic signaling patterns. Therefore, I would recommend this manuscript for publication as long as the following points are addressed:

We thank the reviewer for the careful reading on our manuscript. We also appreciate the reviewer's positive comments on finding the manuscript exciting. Below, we have done our best to address all concerns raised by the reviewer and carefully revised the manuscript accordingly.

1) The authors claim that the method is generally applicable. Although they do test their model on both p53 and MAPK single-cell data and discuss this in the supplemental data I would recommend that the authors should move some of this information/figures to the main text. This would solidify their claim that their approach is generalizable to dynamical signaling systems other than NF- κ B. In addition, although these datasets are published it would be helpful to add labels so that the reader knows what stimuli were used and when without having to refer back to the original papers.

We appreciate the reviewer's great suggestion. We have moved the discussion on p53 and MAPK datasets to the Results section associated with Fig.2. This helps demonstrate that our method of model training is applicable to these datasets. Considering that these datasets did not test multiple stimuli and thus do not allow us to address the central topic of quantifying stimulus discrimination here, we would prefer to keep the figures in SI. The readers could follow the protocol in Fig.S9 and the available code packages to calculate the dMI for new datasets of p53, MAPK, etc, when multiple stimuli are used. Besides, as suggested, we have added the label and timing of the stimuli as the original papers did.

2) The additional signaling datasets raise two other important issues – how many cells and how many timepoints are needed for this approach? Again, the authors do discuss this in the supplement, but there should be an expanded discussion of this in the main text as these are both key decisions a user would need to make to implement the dMI approach described in this paper. It was also unclear how many cells are needed for the time-inhomogeneous Markov model – I may have missed this in the supplement but if not, it should be included.

Following the reviewer's suggestion, we now expand the discussion on the proper numbers of cells and timepoints in the main text. In general, to find the proper numbers of cells and timepoints requires to iterate with the search for the proper number of states by the computational protocol (Fig.S9). For a given dataset with certain numbers of cells and timepoints, one first needs to search for the optimal number of parameters by quantifying the model performance and overfitting. If there is a range of parameters' number to achieve high model performance and avoid overfitting simultaneously (e.g., Fig.S5c), the numbers of cells and timepoints are sufficient. However, if there is a gap to find such optimal number of parameters, it implies to increase the number of measured cells and timepoints. This provides a guidance on whether to measure more cells and timepoints for a given dataset.

For the NFkB dataset with the hidden Markov model, ~500 cells with ~150 timepoints enable us to find such optimal number of parameters with high model performance and avoiding overfitting (Fig.S5c). Under this number of parameters, ~500 cells is required, and overfitting happens if half the number of measured cells are used (Fig.S13). To investigate the required number of timepoints, we performed dMI calculations using less timepoints with lower sampling frequency, where every two time points were used. As shown in the figure here, the dMI in the middle panel has a similar temporal profile as the original sampling frequency (left panel), and is slightly higher. It indicates that ~150 timepoints in 12 hours may be reduced, but reducing half starts to cause overfitting. We further used half the number of states for the subsampled data, which gives a lower dMI (right panel). Then, the optimal number of states for the subsampled data are between the two settings. However, such optimal range of parameters may no longer exist if less timepoints and cells were used. Overall, ~500 cells with ~150 timepoints are sufficient for the NFkB dataset, serving as a reference for other datasets.

As for the time-inhomogeneous Markov model, it can be trained to reproduce the signaling dynamics even with ~ 100 cells, such as shown in Figs.2,S6,S7. However, to avoid overfitting such that the dMI will not be overestimated, the time-inhomogeneous Markov model requires many more cells. The NF κ B, p53, and MAPK datasets do not have sufficient cells, as revealed by the quantification on the overfitting when the time-inhomogeneous Markov is used (Fig.S8): the high model performance typically requires >16 states (Figs.S5-S7), but the overfitting happens when the number of states is just over 4 (Fig.S8b). Thus, there is no optimal number of states that can give high model performance without overfitting, indicating an insufficient number of measured cells. Note that here the dMI for p53 and MAPK datasets does not quantify the stimulus discrimination and is only used to check whether the overfitting happens.

To avoid this overestimation requires at least several fold of 500 cells, because for each two consecutive time points one transition matrix needs to be fitted, whereas the hidden Markov model only requires two matrices in total. Therefore, in practice it is experimentally and computationally expensive to use the time-inhomogeneous Markov model, and the hidden Markov model is generally better for accurate dMI calculations. We have added this comment in the text.

Finally, are the authors able to estimate whether the number of cells needed changes based on how heterogeneous a dataset is in signaling dynamics?

The required number of cells depends on the heterogeneity of the dataset. In general, more measured cells are required for more heterogeneous signaling dynamics, such that the heterogeneous dynamics can be captured by the model. For a given number of cells, one needs to search for the optimal number of parameters that allows high model performance and avoid overfitting, as shown in Fig.S5. When the dataset is more heterogeneous, to achieve high model performance will be harder. Then, the range for “number of states is sufficient to train a model” in Fig.S5c becomes narrower. When there is a gap on the proper number of parameters to have high model performance and avoid overfitting simultaneously, more cells need to be measured, which gives a criterion for the required number of cells. We have provided this guidance in the Discussion.

3) Can the authors comment on whether their choice of 64 hidden and 32 emission states as optimal for the HMM is specific to the NF κ B dataset or do they anticipate that

this is true for other oscillating signaling systems (p53, ERK) or non-oscillating signaling systems (p38)?

The proper number of states may differ across datasets. Specifically, the optimal number of states depends on the number of timepoints, the number of cells, the complexity of the dynamics, and the cell-to-cell heterogeneity. Therefore, we anticipate that the optimal number of states is different for oscillating p53, ERK datasets, especially when the number of measured timepoints and cells differ. The best way to determine the optimal number of states is to follow the computational protocol in Fig.S9, which is applicable to various datasets with distinct dynamical patterns. We also would like to mention that when the optimal number of parameters are found and used for each dataset, the dMI values are comparable across datasets. We now added these points in the Discussion.

Minor points:

Lines 204-205: The authors should add to the text what their conclusion is after exploring how varying the number of hidden or emission states altered HMM performance.

Following the reviewer's suggestion, we now add a short summary on this point after the previous line 205: "Increasing the number of either hidden or emission states generally improves the model performance".

Detailed reply to Reviewer #3

Original reports are typed in blue color and our response in black. Please note the numbers of equations and figures refer to the revised manuscript.

This paper proposes a methodology for quantification of mutual information of time course data. It is interesting, novel and potentially important.

We appreciate the reviewer's careful reading and the positive comments on finding the manuscript interesting, novel and potentially important. Below, we have carefully addressed the reviewer's question and revised the manuscript accordingly.

I have one question to be resolved before publication, though. To what extent does this definition of the mutual information depend the time resolution of the data, and on uncertainties in the data?

The definition of mutual information is the same for all data, regardless of time resolution and uncertainties, but the estimated values do depend on them. To investigate the dependence on the time resolution, we performed dMI calculation using a different sampling frequency, where every second time point was used for the NFkB dataset. As shown in the figure inserted below, the dMI has a similar temporal profile as that when the original sampling frequency was used (left panel). The qualitative behavior of the dMI does not change, and the dMI values of the right panel is slightly higher than the left, indicating the potential overfitting when half of the data points were used. Thus, for the data measured by different time resolution, one may need to search for the optimal parameter by the computational protocol described in Fig.S9.

The estimated mutual information is also affected by uncertainties in the data. When the dataset is more heterogeneous, to achieve high model performance becomes harder. Then, the range for “number of states is sufficient to train a model” in Fig.S5c becomes narrower. When there is a gap between the minimal number of parameters that achieve high model performance and the maximum number of parameters that still avoid overfitting, one needs to measure more cells. We also investigated potential out-of-sample effects by using the replicates measured at different days (Fig.S17). The dMI calculation captures the difference in the replicates, but they have qualitatively similar behaviors and the difference between replicates was not dramatic. We now have added these points in the Discussion.

Detailed reply to Reviewer #4

Original reports are typed in blue color and our response in black. Please note the numbers of equations and figures refer to the revised manuscript, and the numbers of references refer to the list after the reply to each reviewer.

In this manuscript, Tang and co-authors worked on a key question in cell signaling – how do cells encode extracellular inputs as temporal regulator dynamics. To address this question, one would need to quantify the amount of information encoded in the temporal dynamics, and determine how well cells can utilize such encoded information to differentiate different inputs. However, it has been challenging to accurately quantify the dynamics-encoded information. While several methods have been developed to calculate such information, they have not yet fully leveraged the time sequence of the temporal data. In this work, the authors developed a hidden Markov model-based approach to estimate the information encoded in the temporal dynamics, which performed much better compared to existing methods. This novel approach allowed the authors to decipher the temporal changes in the encoded information, and investigate different molecular mechanisms affecting information encoding. Overall, this manuscript is well written and the results are generally nicely presented. What they have revealed provide new insights into how cells encode inputs into temporal signals, and their method should be broadly applicable for understanding dynamic signal processing in different organisms. However, I have several concerns that the authors may wish to address before publication.

We are grateful to the reviewer's careful reading and pithy summary on the major results. We are glad to see that the reviewer finds the manuscript well written and the results insightful and broadly applicable. Below, we have done our best to address the concerns by the reviewer.

Major concerns:

1. A general remark on the data and the MI value. While it is a common practice in the field to quantify signal processing and information encoding with data from multiple different populations of cells, it appears that it should be better to collect the response dynamics from the same population of cells. In other words, the biological definition of “the ability to differentiate different inputs” refers to the ability of a cell to tell one input apart from the other. That said, it makes a lot more sense to calculate MI based on paired datasets, i.e., the dynamic responses of the same set of cells to different inputs. Although there are technical challenges in doing so, working on such paired datasets seems to be more biologically meaningful. It would be great if the authors can elaborate on this point in the manuscript.

We agree that the ideal case would be to measure the signaling responses of the same set of cells to different inputs. In this way, one could reduce the potential heterogeneity among cell populations. In practice, this is not possible as the response to one stimulus will affect the cells to the second stimulus. Therefore, in order to compare signaling

response of the “naive” cells, the experimental strategy is to split the single population of cells into physically separated compartments that are then stimulated with a single stimulus. It is possible that the physical separation causes the subpopulations to be distinct. This is mitigated if the stimulations are done at the same time in parallel on the same microscope run, with the microscope objective gathering the data by moving from compartment to compartment for each timepoint. To address the scenario when experiments cannot be done in parallel, we have investigated how different replicates are (Fig.S17, see more discussion below). In short, while differences are detectable they are minor and the dMI are quite similar. We now elaborate this point in Discussion.

This point is likely related to the generally low MI values reported in the manuscript. For example, in Fig. 3a, the maximum MI is ~ 2 (i.e., corresponds to the ability to differentiate 4 inputs). And for Fig. 4a, the maximum MI is less than 1 (i.e., can't differentiate two inputs). Alternatively, it seems that the MI computed is a “technical” MI rather than the “biological” MI – i.e., cells must have done better than this.

The reviewer mentioned an important point. Indeed, historically the low MI values was a surprising finding in the pioneering work of this direction³. After that, a series of works have attempted to explain the mystery of the low MI values, as one would expect that cells should do better than effectively recognizing only two different stimuli. One important finding was that, if multiple timepoints of the signaling responses is considered, the MI increases¹. Here, we move one step further to reveal that if the dynamics of signaling responses were included, the MI values can continue to accumulate and may reach 2 bits for 13 different conditions ($\log_2 13 \approx 3.7$ bits in the ideal limit) and possibly 1 bit for the pairwise conditions in Fig.4 ($\log_2 2 = 1$ bits in the ideal limit). The loss of the information can be caused by the molecular noise that governs signaling responses. However, we also agree with the reviewer that the measured dMI estimates are always underestimates of the true biological value, because all measurements are subject to technical noise or uncertainty. Indeed, the amount of technical noise remains unknown and no method has been established to reliably distinguish between technical measurement noise and biological heterogeneity.

2. About model training. It is unclear whether the authors used data from all 13 samples to train one model or they used one sample set for training each model (i.e., a total of 13 models). In line 133, the authors said that they train a Markov or hidden Markov model for each stimulus condition, what if training one model for all conditions or for each stimulus type since we can assume that under some conditions the underlying biological networks are the same. This may also reduce the effect of overfitting.

We used **each** of the 13 samples to train one model, and thus obtained 13 trained models in total. Technically, we could train one model for all the conditions or each stimulus type. However, the trained model does not represent the trajectories under a specific stimulus condition, and is not useful to quantify the stimulus discrimination. Note that even for the same receptor ligand type, different concentrations can lead to distinct behaviors of the signaling response, and thus are treated as distinct stimulus conditions. For various stimulus conditions, there can be differences in the abundances of chemical

species in the biological network^{9,12}, and using all conditions together to train one model do not reveal such differences. Therefore, though using all conditions for model training may help reduce overfitting, training one model for each condition is more proper to quantify the stimulus discrimination.

3. About model validation. It appears that the authors used in-sample data for cross-validation. I wonder if there is a way to perform out-of-sample cross-validation with data from biological replicates (that are collected on different days/experiments).

To address the reviewer's question, we now used the data collected from different days to perform an "out-of-sample cross-validation". Specifically, we calculated the dMI for the pairwise stimulus conditions (Fig.4), and used the replicate data to replace one of the two in each pairwise conditions. The replicates were generated at different dates (Table.S3), serving as the out-of-sample validation. As shown in Fig.S17 and attached here, the dMI with using the replicates are similar to Fig.4, validating the robustness of the method.

There are differences in the dMI estimates between the replicates. The deviations are small for the first two pairwise conditions. For the third pairwise stimulus conditions (third row in left panel, red line), though both dMI of the replicates gradually increase in the relatively late phase, the onset time differs. This can be attributed to the fact that the measured data under 100 nM CpG varies between the replicates (which were collected with >1-year gap), as shown by the upper heatmap in the red box of Fig.4 and this figure. In any cases, the dMI robustly reveals such differences. Overall, the out-of-sample measurements may have variations, which are captured by the dMI method.

4. In fig. S7, it is somewhat strange and hard to understand that the HMM model is worse than the time-homogeneous Markov model. Although the authors provided potential explanations, I think the key issue is related to the nature of the data. The data in Fig. S7a are composed of equal segments of pre-stress and post-stress activity data, which is in sharp contrast to their own NFkB datasets in Fig. 2a and Fig. S2a, which are

compose of primarily post-stress activity data. I think this is a key issue because these two types of temporal data are fundamentally different, which likely affects model fitting.

We thank the reviewer for the insightful idea and agree on this point. The previous result that the HMM is worse in the ERK dataset is exactly due to that the data of the two time windows (pre-stress and post-stress) are genuinely different. To validate it, we now perform the model training for the two time windows separately and found that indeed the models can generate similar trajectories to the data, as shown in the figure below. Therefore, the HMM should be able to learn the signaling dynamics under fixed environmental conditions, and distinct HMMs are required when the environment or stimulus-condition is changed.

5. I am curious about the performance of dMI under different sampling frequencies (i.e. different length of interval between time points), is there a minimum time interval for accurate computation of dMI and how is it related to the time scale of the underlying biological process?

We now perform a dMI calculation with using a different sampling frequency and various time windows of data. For the sampling frequency, every two time points were used. As shown in the figure below, the dMI has a similar temporal profile as the original sampling frequency, and is slightly higher than Fig.3a. It indicates that such reduction in the sampling frequency does not change the qualitative behavior of dMI, but starts to cause overfitting and hence overestimations.

For various time windows, we used the first 2,4,6,8,10 hours to do model training the dMI calculations (Fig.S13e and attached below). The dMI is higher when using shorter time course of data, which is caused by overfitting. For the range out of the chosen time window, as shown by transparent lines, the dMI starts to decrease with time. In that time regime, the trajectories are less distinguishable by the model, because the model was not trained by the data of that time regime.

In general, there is no sharp threshold for the minimum time interval for dMI calculation. However, to distinguish various dynamical patterns of signaling response and to get high-resolution on the information accumulation requires the sampling frequency to be sufficiently high and time interval to be long enough. Increasing the number of timepoints would also help reduce the overfitting. As we describe in the text, for each given dataset, one needs to search for the optimal number of parameters by the computational protocol (Fig.S9). For example, here we further use half the number of states for the half-sampling-frequency data (right panel above), where the dMI indicates that the optimal number of states for the subsampled data are between these two settings. Overall, if the timepoints are sufficient for measuring the dynamical patterns and enable accurate model training without overfitting, the dMI can reveal the time scale of information accumulation for the biological process.

6. Line 211-215, the authors chose the best state number for all conditions while under different conditions the underlying biological networks differ. What if choosing the best state number for each condition (even though this may contribute to overfitting)? And in line 241-242, I was confused about the relationship between overfitting and trajectory number. If reducing trajectory number can lead to overfitting, then state number cannot be set the same since the trajectory numbers of different conditions are different.

The primary reason we chose the same number of parameters across conditions is for the convenience of computation of the dMI. Specifically, to calculate conditional trajectory probabilities in Eq.(7) requires to use the model trained by the condition 1's data to calculate the probabilities of condition 2's trajectory. Then, the same number of states allows a direct insertion of condition 2's trajectory to condition 1's hidden Markov model for the calculation. If different state numbers are employed for each condition, more sophisticated procedures need to be designed to match the states.

Different numbers of trajectories for the conditions do affect the overfitting when using the same number of states. However, we found that, though different conditions are differentially prone to overfitting, the variation of overestimated dMI is in the scale 0.1 bit, and does not dramatically affect the result: The error bar in the grey points of Fig.S5b show the standard deviation of the overfitting across conditions. Due to the above reasons, we chose to use the same state number for all the conditions. If one needs to strictly rule out such variations across conditions, another solution would be to use the same number of trajectories for each condition. We now have added this point to Discussion, and thank the reviewer for raising the point.

7. Biological insights. The authors said that stimuli distinction occurs in different temporal orders. I wonder if these temporal orders are related to signaling cascades in cells. Can the authors combine their results with related signaling pathways to further prove their methods and reveal more biological insights? Is there any difference for temporal profiles of dMI between within-stress-type comparison and across-stress-type comparison? If so, is it related to underlying signaling networks? And in Line 294, can the authors use the 13-sample data to prove these hypotheses?

The reviewer raised interesting questions. As speculated in Fig.5, the temporal orders of stimuli discrimination depend on molecular circuit motifs or mechanisms within the signaling network. In particular, we used the I κ B-mutant data to show that the negative feedback loop is responsible for encoding the information in the middle phase of 1-2 hours via oscillatory patterns. On the other hand, the result from the 13-conditions data of WT cells does not contain genetic perturbations of the signaling network, and thus is not the best to address the hypotheses in the previous line 294. Even though one could classify the 13-conditions into within-stress-type and across-stress-type, it lacks a direct connection to signaling pathways.

To further investigate the role of signaling network on the dMI, we now use simulated datasets from an established mathematical model of the NF κ B signaling network^{11,12}. We employed the model to generate data for dMI calculations, as shown below. We perturbed key parameters of the signaling network, which affect the early activation, intermediate-phase oscillation, and late sustained features separately. Then, we applied the data to the established workflow and calculated the dMI for specific pairwise conditions, which have perturbations on distinct parameters of the signaling network as denoted on top of each heatmap. The dMI shows distinct temporal profiles corresponding to the difference on certain dynamical features. For example, for the perturbation of the early activation mechanism by modifying the ligand-receptor

interaction (heatmaps 1,2), the dMI increases in the early phase. For the perturbation on the intermediate-phase oscillation by altering the negative feedback of I κ B α (heatmaps 3,4), the dMI increases when the oscillation patterns start to have difference around 1 hour. For the perturbation on the features of duration and integral by changing the stimulus life-time (heatmaps 1,5), the dMI increases relatively late, around 2 hours. Note that there may be no clear-cut separation on the features as the source of dMI, such as that the dMI can be attributed to both oscillation and duration for the third pair.

Therefore, the application of dMI to the simulated trajectories from the mathematical signaling model help demonstrate potential strategies to perturb the signaling network that can alter the temporal information transmission. It serves as a computational exploration on the hypothesis in the previous line 294, beyond only using the 13-conditions data without genetic perturbation. Further experiments to conduct genetic perturbation are required to validate the hypothesis.

To explore more biological insights, we also analyzed the data of NFκB-responsive genes published in ⁹ by our lab, and investigated the consequence of dMI of NFκB signaling to downstream gene expressions. With the measured NFκB-responsive gene expression at three time points (1, 3, 8 hours) under the treatment of TNF, Poly(I:C), or CpG separately, we calculated the correlation between the dMI values and the absolute difference of gene expression fold change between the pairwise stimuli. From the histogram of this correlation (panel b), a large proportional of genes track the signaling information of NFκB. More time points of expression measurement may be required to further validate the correlation.

We further chose representative genes whose expression patterns appear to follow the dMI: the large absolute differences of gene expression fold change between the pairwise stimuli correlate with high dMI values, and vice versa. For example, the protease inhibitor Serpina3f ¹⁰ is induced when macrophages sense bacteria (LPS) locking up infected tissues to prevent bacterial spread, but not when they sense TNF which derives from neighboring cells. The distinction is sensed early and is sustained, which matches with the dMI between LPS and TNF. See more descriptions on the genes and discussions in SI Section IV.D.1. These results help demonstrate the biological insights from the dMI.

Minor points:

1. Need to proofread the manuscript more carefully, e.g., Line 41 etc

We thank the reviewer for the suggestions and have revised “plays” to “play” in the previous line 41. We have also proofread the manuscript again to revise the typos.

2. Line 355, is the normalization term wrong (if normalized by rows)? What's more, since the meaning of p has changed after normalization, this symbol should be changed after normalization.

We have checked the normalization in the previous line 355, and changed it to “The transition matrix was obtained by normalizing the count matrix by each column”. As suggested by the reviewer, we have also revised the symbol of joint count matrix to be $c(y_n, y_{n-1})|_{t=n-1}$, and used p only after the normalization.

3. In the legend of Fig. S16a, “Fig. 5's panel (c)” should be “Fig. 5's panel (b)”.

We have changed “Fig.5's panel (c)” to “Fig.5's panel (b)”. Thanks again for the careful reading.

References:

1. Selimkhanov, J. *et al.* Accurate information transmission through dynamic biochemical signaling networks. *Science* **346**, 1370–1373 (2014).
2. Seifert, U. Entropy Production along a Stochastic Trajectory and an Integral Fluctuation Theorem. *Phys. Rev. Lett.* **95**, 040602 (2005).
3. Cheong, R., Rhee, A., Wang, C. J., Nemenman, I. & Levchenko, A. Information Transduction Capacity of Noisy Biochemical Signaling Networks. *Science* **334**, 354–358 (2011).
4. Chen, S. Y. *et al.* Optogenetic Control Reveals Differential Promoter Interpretation of Transcription Factor Nuclear Translocation Dynamics. *Cell Systems* **11**, 336-353.e24 (2020).
5. Hao, N. & O’Shea, E. K. Signal-dependent dynamics of transcription factor translocation controls gene expression. *Nature Structural & Molecular Biology* **19**, 31–39 (2012).
6. Purvis, J. E. & Lahav, G. Encoding and Decoding Cellular Information through Signaling Dynamics. *Cell* **152**, 945–956 (2013).
7. Sen, S., Cheng, Z., Sheu, K. M., Chen, Y. H. & Hoffmann, A. Gene Regulatory Strategies that Decode the Duration of NFκB Dynamics Contribute to LPS- versus TNF-Specific Gene Expression. *Cell Systems* **10**, 169-182.e5 (2020).
8. Granados, A. A. *et al.* Distributed and dynamic intracellular organization of extracellular information. *PNAS* **115**, 6088–6093 (2018).
9. Cheng, C. S. *et al.* Iterative Modeling Reveals Evidence of Sequential Transcriptional Control Mechanisms. *Cell Systems* **4**, 330-343.e5 (2017).
10. Heit, C. *et al.* Update of the human and mouse SERPINE gene superfamily. *Human Genomics* **7**, 22 (2013).
11. Taylor, B., Adelaja, A., Liu, Y., Luecke, S. & Hoffmann, A. Identification and physiological significance of temporal NFκB signaling codewords deployed by macrophages to classify immune threats. *bioRxiv* 2020.05.23.112862 (2020).
12. Maity, A. & Wollman, R. Information transmission from NFκB signaling dynamics to gene expression. *PLOS Computational Biology* **16**, e1008011 (2020).

Reviewers' Comments:

Reviewer #1:

Remarks to the Author:

I have read the revised manuscript and the rebuttal letter carefully and concluded that the authors have satisfactorily addressed all my major concerns. I remain enthusiastic about the findings and impact of the work, and support publication of the revised manuscript in Nature Communications.

Reviewer #3:

Remarks to the Author:

The authors have satisfactorily addressed all of my comments, and I appreciate their detailed responses and additions to the revised manuscript. I have no further comments or questions.

Reviewer #4:

Remarks to the Author:

My issues have been addressed. I recommend publication.

Reviewer #5:

Remarks to the Author:

The authors have done a superb job in addressing the comments. They have now more thoroughly discussed the limits and strengths of their approach. I recommend the publication of the manuscript.